# DTGB: A Comprehensive Benchmark for Dynamic Text-Attributed Graphs

Jiasheng Zhang[1]    Jialin Chen[2]    Menglin Yang[2]    Aosong Feng[2]    Shuang Liang[1]
Jie Shao[1,3]*    Rex Ying[2]

[1]University of Electronic Science and Technology of China    [2]Yale University
[3]Shenzhen Institute for Advanced Study, University of Electronic Science and Technology of China
zjss12358@std.uestc.edu.cn   {shuangliang, shaojie}@uestc.edu.cn
{jialin.chen, menglin.yang, aosong.feng, rex.ying}@yale.edu

## Abstract

Dynamic text-attributed graphs (DyTAGs) are prevalent in various real-world scenarios, where each node and edge are associated with text descriptions, and both the graph structure and text descriptions evolve over time. Despite their broad applicability, there is a notable scarcity of benchmark datasets tailored to DyTAGs, which hinders the potential advancement in many research fields. To address this gap, we introduce **D**ynamic **T**ext-attributed **G**raph **B**enchmark (**DTGB**), a collection of large-scale, time-evolving graphs from diverse domains, with nodes and edges enriched by dynamically changing text attributes and categories. To facilitate the use of DTGB, we design standardized evaluation procedures based on four real-world use cases: future link prediction, destination node retrieval, edge classification, and textual relation generation. These tasks require models to understand both dynamic graph structures and natural language, highlighting the unique challenges posed by DyTAGs. Moreover, we conduct extensive benchmark experiments on DTGB, evaluating 7 popular dynamic graph learning algorithms and their variants of adapting to text attributes with LLM embeddings, along with 6 powerful large language models (LLMs). Our results show the limitations of existing models in handling DyTAGs. Our analysis also demonstrates the utility of DTGB in investigating the incorporation of structural and textual dynamics. The proposed DTGB fosters research on DyTAGs and their broad applications. It offers a comprehensive benchmark for evaluating and advancing models to handle the interplay between dynamic graph structures and natural language. The dataset and source code are available at `https://github.com/zjs123/DTGB`.

## 1   Introduction

Dynamic graphs are an essential tool for modeling a wide range of real-world systems, such as e-commerce platforms, social networks, and knowledge graphs [1, 2, 3, 4, 5, 6, 7]. In those dynamic graphs, nodes and edges are typically associated with text attributes, giving rise to dynamic text-attributed graphs (DyTAGs). For example, e-commerce graphs may contain items accompanied by textual descriptions, and time-annotated edges representing user reviews or interactions with items. Similarly, temporal knowledge graphs represent the sequential interactions among real-world entities through textural relations. The exploration of learning methodologies applied to DyTAGs is important to research areas such as dynamic graph modeling and natural language processing [8, 9, 10], as well as various real-world applications, *e.g.,* recommendation and social analysis [3, 11, 12, 13, 14].

---

*Corresponding author.

38th Conference on Neural Information Processing Systems (NeurIPS 2024) Track on Datasets and Benchmarks.

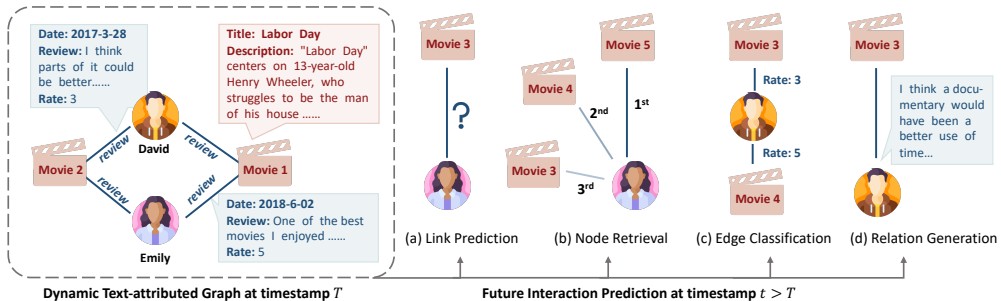

Figure 1: Dynamic text-attributed graph and evaluation tasks: a case study with movie reviews. Given a DyTAG with interactions before timestamp $T$, the tasks are to forecast future interactions in $t > T$, as well as their detailed interaction types and textual descriptions.

While previous works for dynamic graph learning have proposed many datasets describing the temporal interactions across different domains [15, 16, 9, 17], these datasets often lack edge categories and only contain statistical features derived from raw attributes, lacking the raw text descriptions of nodes and edges. Therefore, they fall short in facilitating methodological advances in semantic modeling within dynamic graphs and exploring the impact of text attributes on downstream tasks. Concurrently, text-attributed graphs (TAGs) are widely used in many real-world scenarios [18, 19, 20, 21]. The recently proposed CS-TAG benchmark dataset [22] with rich raw text aims to facilitate research in TAG analysis. However, these datasets oversimplify the evolving interactions in the real world by representing them as static graphs, ignoring the inherent temporal information present in real-world TAG scenarios, such as citation networks with timestamped publications and social networks with chronological user interactions. Consequently, there is an urgent need for benchmark datasets that can accurately capture both dynamic graph structures and rich text attributes of DyTAGs.

**Proposed Work**. To address this gap, we introduce a comprehensive benchmark DTGB, which comprises eight large-scale DyTAGs sourced from diverse domains including e-commerce, social networks, multi-round dialogue, and knowledge graphs. Nodes and edges in the DTGB dataset are associated with rich text descriptions and edges are annotated with meaningful categories. The dataset construction involves a meticulous process, including the selection of data sources, the construction of the graph structure, and the extraction of text and category information (detailed in Section 3). Compared with existing datasets [22, 23, 15, 24], which either lack raw text and temporal annotations or are small in scale and devoid of long-term dynamic structures, DTGB distinguishes itself with its *rich text*, *long-range historical information*, and *large-scale dynamic structures*, ensuring diverse and representative samples of real-world DyTAG scenarios.

With DTGB, we design four critical downstream evaluation tasks based on real-world use cases, as shown in Figure 1. Except for the widely studied future link prediction task, we delve deeper into three more interesting and challenging tasks: destination node retrieval, edge classification, and textural relation generation, which are neglected by previous works. These tasks are fundamental to real-world applications that require the model to handle the temporal evolution of both graph structure and textual information. Our benchmarking results on 7 dynamic graph learning algorithms and 6 LLMs highlight these challenges and showcase the performance and limitations of current algorithms (*e.g.,* the scalability issue of memory-based models, neglect of the edge modeling, and weakness in capturing long-range and semantic relevance), offering valuable insights into integrating structural and textual dynamics. Our **contributions** are summarized as follows:

- **First DyTAG Benchmark.** To the best of our knowledge, DTGB is the first open benchmark specifically designed for dynamic text-attributed graphs. We collect eight DyTAG datasets from a wide range of domains and organize them in a unified structure, providing a comprehensive testbed for model evaluation in this area.
- **Standardized Evaluation Protocol.** We design four critical downstream tasks and standardize the evaluation process with DTGB. This comprehensive evaluation highlights the unique challenges posed by DyTAGs and demonstrates the utility of the dataset for assessing algorithm performance.
- **Empirical Observation.** Our experimental results demonstrate that rich textural information consistently enhances downstream graph learning, such as destination node retrieval and

edge classification. This contributes to a deeper understanding of the complexities involved in DyTAGs and offers guidance for future research in this field.

## 2 Related Works

**Dynamic Graph Learning**. Deep learning on dynamic graphs has gained significant attention in various domains such as social networks, transportation systems, and biological networks [25, 26, 27, 28, 29]. Previous works have proposed a few real-world datasets [9, 16, 30, 31, 32], which offer a comprehensive collection of temporal interaction data across different contexts. Benchmark frameworks [33, 34, 8, 35], such as DyGLib [8], have been instrumental in standardizing the evaluation of dynamic graph models, offering robust metrics for assessing downstream performance. Recent advancements in temporal graph models have significantly enhanced the ability to capture time-evolving relationships in graph-structured data [36, 37, 38, 39, 40, 41, 42, 17], leading to state-of-the-art performance in various tasks such as dynamic link prediction and node classification on temporal graphs. However, existing temporal graph datasets may lack node or edge attributes, or contain simple node/edge features based on bag-of-words [43] or word2vec [44] algorithms derived from the associated text, which are limited in capturing the complicated semantics of the text. In this work, we focus on dynamic graphs where nodes and edges are associated with raw textual descriptions, enabling richer, context-aware representations and more sophisticated downstream tasks.

**Text-attributed Graph Learning**. Text-attributed graphs (TAG) are widely used in various real-world applications. For example, in citation networks, the text associated with each article provides valuable information such as abstracts and titles [45, 18, 46]. CS-TAG [22] offers standardized datasets with raw text, facilitating research and methodological advances in TAG analysis. Recently, large language models (LLMs) have demonstrated remarkable capabilities in enhancing feature encoding and node classification on TAGs [47, 48, 49]. By flattening graph structures and associated textual information into prompts [50, 10, 51], LLMs can leverage their strong language understanding and generation abilities to improve TAG analysis tasks. However, all these TAGs eliminate the temporal information within the graphs, which is inherent and crucial in real-world scenarios. There has been limited exploration of temporal relations in TAGs, which represents a missed opportunity to evaluate the temporal awareness and reasoning ability of LLMs.

## 3 Dataset Details

**Motivation of DTGB**. To investigate the necessity of a comprehensive benchmark dataset for dynamic text-attributed graphs, we first survey various dynamic graphs and text-attributed graphs previously utilized in the literature. We observed that most commonly used dynamic graphs are essentially text-attributed. Simultaneously, many popular text-attributed graphs have inherent temporal information. For instance, the well-known dynamic graph dataset `tgbl-review` [16] and the commonly used TAG dataset `Books-Children` [22] are both derived from the Amazon product review network [52] which is intrinsically associated with both the text attributes of users and products and the time annotations of user-item interactions. However, `tgbl-review` only contains numerical attributes derived from the raw text and `Books-Children` ignores all temporal information, which highly limits the full exploration of the performance for downstream tasks.

While these previous datasets are frequently used, they possess obvious inadequacies when exploring the effectiveness of dynamic graph learning methods in handling real-world scenarios. Firstly, existing dynamic graph datasets lack the availability of raw textual information, bringing challenges to investigating the benefits of text attribute modeling on real-world applications. Secondly, most existing dynamic graph datasets lack reasonable temporal segmentation and aggregation, making their edges distribution quite sparse in the time dimension (*e.g.,* `MOOC` [15] and `tgbl-wiki` [16]). This brings challenges to investigating the structure dependency and evolution for dynamic graphs. Lastly, although TAGs are enriched with node text attributes, they tend to miss edge text and time annotations, making them fail to faithfully reflect the challenges in modeling real-world scenarios.

**Dataset Construction**. To address these challenges, we collect resources from different domains and follow a rigorous process to construct the comprehensive benchmark dataset DTGB for DyTAGs. We carefully select data sources from various domains to ensure diversity and relevance. For graph

Table 1: Statistics of datasets and comparison with existing datasets.

| | Dataset | Nodes | Edges | Edge Categories | Timestamps | Domain | Text Attributes | Bipartite Graph |
|---|---|---|---|---|---|---|---|---|
| Previous Dynamic Graphs | **tgbn-trade** | 255 | 468,245 | N.A. | 32 | Trade | ✗ | ✗ |
| | **tgbl-wiki** | 9,227 | 157,474 | N.A. | 152,757 | Interaction | ✗ | ✗ |
| | **tgbl-review** | 352,637 | 4,873,540 | N.A. | 6,865 | E-commerce | ✗ | ✓ |
| | **MOOC** | 7,144 | 411,749 | N.A. | 345,600 | Interaction | ✗ | ✓ |
| | **LastFM** | 1,980 | 1,293,103 | N.A. | 1,283,614 | Interaction | ✗ | ✓ |
| Previous TAGs | **ogbn-arxiv-TA** | 169,343 | 1,166,243 | N.A. | N.A. | Academic | Node | ✗ |
| | **CitationV8** | 1,106,759 | 6,120,897 | N.A. | N.A. | Academic | Node | ✗ |
| | **Books-Children** | 76,875 | 1,554,578 | N.A. | N.A. | E-commerce | Node | ✗ |
| | **Ele-Computers** | 87,229 | 721,081 | N.A. | N.A. | E-commerce | Node | ✗ |
| | **Sports-Fitness** | 173,055 | 1,773,500 | N.A. | N.A. | E-commerce | Node | ✗ |
| Ours | **Enron** | 42,711 | 797,907 | 10 | 1,006 | E-mail | Node & Edge | ✗ |
| | **GDELT** | 6,786 | 1,339,245 | 237 | 2,591 | Knowledge graph | Node & Edge | ✗ |
| | **ICEWS1819** | 31,796 | 1,100,071 | 266 | 730 | Knowledge graph | Node & Edge | ✗ |
| | **Stack elec** | 397,702 | 1,262,225 | 2 | 5,224 | Multi-round dialogue | Node & Edge | ✓ |
| | **Stack ubuntu** | 674,248 | 1,497,006 | 2 | 4,972 | Multi-round dialogue | Node & Edge | ✓ |
| | **Googlemap CT** | 111,168 | 1,380,623 | 5 | 55,521 | E-commerce | Node & Edge | ✓ |
| | **Amazon movies** | 293,566 | 3,217,324 | 5 | 7,287 | E-commerce | Node & Edge | ✓ |
| | **Yelp** | 2,138,242 | 6,990,189 | 5 | 6,036 | E-commerce | Node & Edge | ✓ |

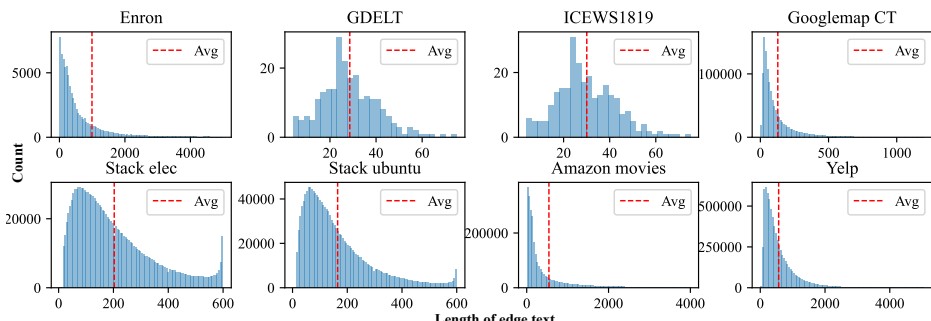

Figure 2: Distribution of edge text lengths on the DTGB datasets.

construction, the redundant and low-quality records are first filtered out and we divide the data into discrete time intervals. In each time interval, nodes and edges are flexibly identified for different domains. Nodes could represent users, products, questions, *etc.* while edges represent relationships such as transactions and reviews. For text and category extraction, we organize multiple text descriptions from the source data and remove the meaningless or garbled characters and low-quality text. We categorize edges based on predefined criteria relevant to real-world use cases such as product ratings and content topics. This process ensures that the dataset accurately reflects the complexity of real-world DyTAG. Taking `Googlemap CT` and `Amazon movies` as an example, which are extracted from Recommender Systems and Personalization Datasets [53]. Nodes represent users or items, while edges indicate review relation between users and items. The original data is first reduced to a $k$-core subgraph, indicating that each user or item has at least $k$ reviews. Edges are segmented by days and edges within each day are aggregated as a subgraph. Edge categories are integers from 1 to 5, derived from the ratings from users to items. Node text includes the basic information of the item (*e.g.,* name, description, and category). Edge text is the raw review from users. Detailed descriptions of all the datasets can be found in Appendix A.1.

**Distribution and Statistics**. Table 1 gives the statistics of previous datasets and DTGB datasets. One can notice that compared with previous dynamic graph datasets, our datasets are characterized by edge categories and text attributes at both node and edge levels. Our dataset includes small, medium, and large graphs with various distributions from four different domains, encompassing both bipartite and non-bipartite, long-range and short-range dynamic graphs. We further study the data distribution to better understand our benchmark datasets. As shown in Figure 2 and Figure 3, datasets from the same domain exhibit similar distributions. For example, knowledge graph datasets `GDELT` and `ICEWS1819` both approximate Gaussian distributions in edge text length, and e-commerce datasets `Googlemap CT`, `Amazon movies`, and `Yelp` show long-tail distributions in the number of edges per timestamp. This demonstrates that our datasets have faithfully preserved the characteristics of data from different domains. More detailed analysis of our datasets can be found in Appendix A.2.

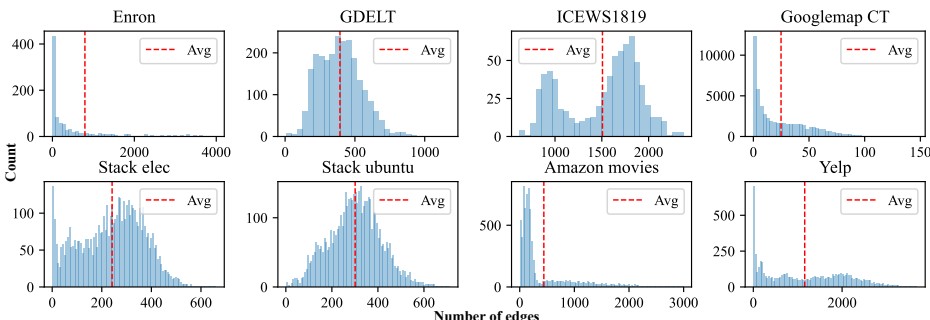

Figure 3: Distribution of the numbers of edges in each timestamp on the DTGB datasets.

# 4 Formulation and Benchmarking Tasks on DyTAG

**DyTAG Formulation**. A DyTAG can be defined as $\mathcal{G} = \{\mathcal{V}, \mathcal{E}\}$, where $\mathcal{V}$ is the node set, $\mathcal{E} \subset \mathcal{V} \times \mathcal{V}$ is the edge set. Let $\mathcal{T}$ denote the set of observed timestamps, $\mathcal{D}$, $\mathcal{R}$ and $\mathcal{L}$ are the set of node text descriptions, edge text descriptions, and edge categories, respectively. Each $v \in \mathcal{V}$ is associated with a text description $d_v \in \mathcal{D}$. Each $(u, v) \in \mathcal{E}$ can be represented as $(r_{u,v}, l_{u,v}, t_{u,v})$ with a text description $r_{u,v} \in \mathcal{R}$, a category $l_{u,v} \in \mathcal{L}$ and a timestamp $t_{u,v} \in \mathcal{T}$ to indicate the occurring time of this edge. We use $\mathcal{G}_T = \{\mathcal{V}_T, \mathcal{E}_T\}$ to represent the DyTAG containing interactions occurred before timestamp $T$. We summarize the important notations used in Appendix B.

**Future Link Prediction**. Future link prediction is commonly used in previous literature [16, 8, 9] to evaluate the performance of dynamic graph learning methods, which aims to predict whether two nodes will be linked in the future given the history edges. In dynamic text-attributed graphs, the new linkage between nodes not only depends on their static semantics brought by node text but also on the interaction context brought by edge text semantics. The future link prediction task in the DyTAG setting can be viewed as the simplification of many real-world applications, *e.g.,* predicting whether one person will e-mail another based on the content of their history e-mails. Formally, given the DyTAG $\mathcal{G}_T$ which contains edges before timestamp $T$, future link prediction aims to predict whether an interaction will happen between nodes $u$ and $v$ at timestamp $T + 1$. In the inductive setting, either $u$ or $v$ are new nodes not contained in $\mathcal{V}_T$.

**Destination Node Retrieval**. While future link prediction has been widely used, this task has several limitations for a reliable evaluation. Its performance largely depends on the quality and the size of the sampled negative samples, and the binary classification metrics are sensitive to the model fluctuations. Destination node retrieval is a novel task that aims to rank the most likely interact nodes for a given node based on its interaction history. This approach is more stable as it considers the relative relevance among the entire node set and is more applicable to real-world scenarios (*e.g.,* personalized recommendation). Formally, given node $u$ and $\mathcal{G}_T$, node retrieval aims to rank the nodes in $\mathcal{V}_T$ based on their possibilities of interacting with $u$ in timestamp $T + 1$. In the inductive setting, $u$ is a new node not contained in $\mathcal{V}_T$.

**Edge Classification**. Edge classification is an essential evaluation task for DyTAG, which is under-explored by previous dynamic graph benchmarks [16, 8, 9]. The dynamic graph learning models leverage both rich textual information and historical interactions to predict the categories of relations between them (*e.g.,* the review rating in the future). Formally, given a DyTAG $\mathcal{G}_T$ with edges up to timestamp $T$, edge classification aims to predict the category of a potential edge at timestamp $T + 1$, utilizing both node and edge textual attributes and historical interactions.

**Textural Relation Generation**. Textural relation generation is a novel task that seeks to leverage historical interactions and their associated text to generate future relation context. While previous studies on TAGs [22, 48] have mainly focused on graph structure learning, such as predicting new edges or node labels, generating the actual textual content for future edges remains an under-explored challenge. To address this gap, large language models (LLMs) are employed as backbones, due to their powerful capability in understanding and generating natural language. Specifically, given two nodes $u$ and $v$ for which we aim to predict their future textual interaction, we provide the LLM with their node descriptions as well as the historical one-hop interactions involving either $u$ or $v$. We then

prompt the LLM to generate the predicted interaction text in an autoregressive manner. This task not only serves as a challenging benchmark for evaluating LLMs' ability to understand the co-evolution of graph structures and natural language, but also holds promise for enhancing LLMs' representations by incorporating the inductive biases of structured data during pretraining in the future.

## 5 Experiments

**Baselines**. (1) For the future link prediction, destination node retrieval, and edge classification tasks, we use the dynamic graph learning models as the baselines. We evaluate 7 popular and state-of-the-art models: JODIE [54], DyRep [42], TGAT [37], CAWN [39], TCL [41], GraphMixer [40] and DyGFormer [8]. (2) For the textual relation generation task, we benchmark four open-source large language models: Mistral 7B [55], Vicuna 7B/13B [56], Llama-3 8B [57], and two closed-source large language models GPT3.5-turbo and the most recent GPT4o through API service. Refer to Appendix C.1 for more details.

**Evaluation Metrics**. For the edge classification task, we use Weighted Precision, Weighted Recall, and Weighted F1 score to evaluate the model performance. For the future link prediction task, we follow previous works [16, 8] and adopt the Average Precision (AP) and Area Under the Receiver Operating Characteristic Curve (AUC-ROC) as the evaluation metrics. For the node retrieval task, we use Hits@$k$ as the metric, which reports whether the correct item appears within the top-$k$ results generated by the model. To evaluate the generated textual relation, we use BERTScore [58], which leverages a pre-trained language model and calculates the cosine similarity between the prediction and ground truth. The detailed definitions are provided in Appendix C.2.

**Experimental Settings**. For dynamic graph learning models, we follow the implementations from DyGLib[1] [8]. All data loading, training, and evaluation processes are performed uniformly, following DyGLib. To integrate the textual information, we use the Bert-base-uncased model [59] to encode the node and edge texts as the initialization of the node and edge representations. We chronologically split each dataset into train/validation/test sets by 7:1.5:1.5. For the textual relation generation task, open-source LLMs are implemented with Huggingface [60]. We also use the parameter-efficient fine-tuning method, LoRA [61], to fine-tune the $\mathbf{Q}$, $\mathbf{K}$, $\mathbf{V}$, and $\mathbf{O}$ matrices within LLM for better text generation. We run all the models five times with different seeds and report the average performance to eliminate deviations. Experiments are conducted on NVIDIA A40 with 48 GB memory.

**Implementation Details of Dynamic Graph Models**. For all of the edge classification task, future link prediction task, and destination node retrieval task, we use Adam [62] for optimization. All the models are trained for 500 epochs and use the early stopping strategy with a patience of 5. The batch size is set as 256. For the edge classification task, after obtaining the representations of the target node and source node, we feed the concatenated representations of two nodes into a multi-layer perceptron to perform the multi-class classification. We employ the cross-entropy loss function to supervise the training in this task. The future link prediction task and the destination node retrieval task share the same training process where a multi-layer perception is used to obtain the possibility scores and the binary cross-entropy loss function is used for supervision. After obtaining the possibility scores of test samples, we traverse different thresholds to get the AP and AUC-ROC metrics for the future link prediction task, and rank the possibility scores of all candidates to get the Hits@k metric for the destination node retrieval task. We perform the grid search based on the performance of the validation set to find the best settings of some critical hyperparameters, where the searched ranges and related models are shown in Table 2. We use the vanilla recurrent neural network as the memory updater of JODIE and DyRep. For CAWN, the time scaling factor is set as $1e - 6$, and the length of each walk is set as 2. To integrate the text information, we use the pre-trained language model (*i.e.,* Bert-base-uncased[2]) to get the representations of text attributes, and then these representations are used to initialize the embeddings of nodes and edges in models. The dimensions of the pre-trained representations are 768 and the maximum text length is set as 512.

**Implementation Details of Large Language Models**. For the textual relation generation task, the inputs to LLMs include the text attribute of the source node and target node, the recent $k$ edges from the source node and the corresponding text, and the recent $k$ edges from the target node and

---

[1] https://github.com/yule-BUAA/DyGLib
[2] https://huggingface.co/google-bert/bert-base-uncased

Table 2: Searched ranges of hyperparameters and the related dynamic graph learning models.

| Hyperparameters | Searched Ranges | Related Models |
|---|---|---|
| Dropout Rate | [0.0, 0.2, 0.4, 0.6] | All of 7 models |
| Sampling Size | [10, 20, 30] | DyRep, TGAT, TCL, GraphMixer |
| Learning Rate | [0.0001, 0.0005, 0.001] | All of 7 models |
| Sampling Strategies | [uniform,recent] | TCL, GraphMixer |
| Number of Walks | [16, 32, 64, 128] | CAWN |
| Sequence Length | [32, 64, 128, 256, 512, 1024, 2048, 4096] | DyGFormer |
| Patch Size | [1, 2, 4, 8, 16, 32, 64, 128] | DyGFormer |
| Number of GNN Layers | [1, 2, 3] | DyRep, TGAT |
| Number of Transformer Layers | [1, 2, 3] | TCL, DyGFormer |
| Number of Attention Heads | [2, 4, 6, 8] | DyRep, TGAT, CAWN, TCL, DyGFormer |

Table 3: Performance of dynamic graph learning methods in the edge classification task when using Bert-encoded embeddings to initialize the model representations. OOM means out-of-memory.

| Datasets | Models | JODIE | DyRep | TGAT | CAWN | TCL | GraphMixer | DyGFormer |
|---|---|---|---|---|---|---|---|---|
| **Enron** | Precision | $0.6568 \pm 0.0043$ | $\mathbf{0.6635 \pm 0.0052}$ | $0.6148 \pm 0.0012$ | $0.6076 \pm 0.0070$ | $0.5530 \pm 0.0079$ | $0.6313 \pm 0.0024$ | $0.6601 \pm 0.0067$ |
| | Recall | $\mathbf{0.6472 \pm 0.0039}$ | $0.6390 \pm 0.0089$ | $0.5530 \pm 0.0001$ | $0.5783 \pm 0.0094$ | $0.5394 \pm 0.0061$ | $0.5735 \pm 0.0015$ | $0.5802 \pm 0.0071$ |
| | F1 | $\mathbf{0.6478 \pm 0.0065}$ | $0.6432 \pm 0.0062$ | $0.5519 \pm 0.0028$ | $0.5685 \pm 0.0132$ | $0.5177 \pm 0.0044$ | $0.5507 \pm 0.0019$ | $0.5604 \pm 0.0063$ |
| **GDELT** | Precision | $0.1361 \pm 0.0036$ | $0.1451 \pm 0.0071$ | $0.1241 \pm 0.0056$ | $\mathbf{0.1781 \pm 0.0011}$ | $0.1229 \pm 0.0021$ | $0.1293 \pm 0.0026$ | $0.1775 \pm 0.0041$ |
| | Recall | $0.1338 \pm 0.0013$ | $0.1365 \pm 0.0013$ | $0.1321 \pm 0.0042$ | $0.1545 \pm 0.0001$ | $0.1235 \pm 0.0047$ | $0.1320 \pm 0.0008$ | $\mathbf{0.1580 \pm 0.0052}$ |
| | F1 | $0.0992 \pm 0.0009$ | $0.1039 \pm 0.0012$ | $0.0967 \pm 0.0010$ | $\mathbf{0.1340 \pm 0.0012}$ | $0.0987 \pm 0.0051$ | $0.1014 \pm 0.0017$ | $0.1291 \pm 0.0068$ |
| **ICEWS1819** | Precision | $0.3106 \pm 0.0023$ | $0.3270 \pm 0.0025$ | $0.3013 \pm 0.0007$ | $\mathbf{0.3451 \pm 0.0023}$ | $0.3212 \pm 0.0096$ | $0.2999 \pm 0.0022$ | $0.3297 \pm 0.0034$ |
| | Recall | $0.3494 \pm 0.0018$ | $0.3636 \pm 0.0020$ | $0.3512 \pm 0.0006$ | $\mathbf{0.3676 \pm 0.0034}$ | $0.3517 \pm 0.0009$ | $0.3502 \pm 0.0001$ | $0.3632 \pm 0.0026$ |
| | F1 | $0.2965 \pm 0.0008$ | $0.3097 \pm 0.0006$ | $0.2908 \pm 0.0008$ | $\mathbf{0.3156 \pm 0.0057}$ | $0.2939 \pm 0.0022$ | $0.2903 \pm 0.0008$ | $0.3079 \pm 0.0027$ |
| **Googlemap CT** | Precision | $0.6163 \pm 0.0032$ | $0.6073 \pm 0.0019$ | $0.6160 \pm 0.0001$ | $0.6166 \pm 0.0023$ | $\mathbf{0.6213 \pm 0.0087}$ | $0.6171 \pm 0.0020$ | $0.6166 \pm 0.0003$ |
| | Recall | $0.6871 \pm 0.0002$ | $0.6827 \pm 0.0006$ | $0.6862 \pm 0.0002$ | $0.6870 \pm 0.0001$ | $0.6875 \pm 0.0001$ | $0.6872 \pm 0.0003$ | $\mathbf{0.6877 \pm 0.0002}$ |
| | F1 | $0.6189 \pm 0.0016$ | $0.6134 \pm 0.0006$ | $0.6225 \pm 0.0015$ | $0.6187 \pm 0.0003$ | $\mathbf{0.6230 \pm 0.0003}$ | $0.6185 \pm 0.0005$ | $0.6196 \pm 0.0008$ |
| **Stack elec** | Precision | OOM | OOM | $0.6265 \pm 0.0046$ | $0.6167 \pm 0.0094$ | $\mathbf{0.6325 \pm 0.0023}$ | $0.6074 \pm 0.0039$ | $0.6026 \pm 0.0471$ |
| | Recall | OOM | OOM | $0.7205 \pm 0.0094$ | $0.6313 \pm 0.0462$ | $\mathbf{0.7474 \pm 0.0004}$ | $0.7412 \pm 0.0061$ | $0.5891 \pm 0.2747$ |
| | F1 | OOM | OOM | $\mathbf{0.6496 \pm 0.0032}$ | $0.6209 \pm 0.0216$ | $0.6420 \pm 0.0003$ | $0.6412 \pm 0.0005$ | $0.4860 \pm 0.2686$ |
| **Stack ubuntu** | Precision | OOM | OOM | $0.6858 \pm 0.0047$ | $0.6921 \pm 0.0040$ | $0.6915 \pm 0.0118$ | $\mathbf{0.6930 \pm 0.0028}$ | $0.6789 \pm 0.0490$ |
| | Recall | OOM | OOM | $\mathbf{0.7921 \pm 0.0012}$ | $0.5650 \pm 0.1015$ | $0.7880 \pm 0.0026$ | $0.7902 \pm 0.0130$ | $0.7494 \pm 0.0991$ |
| | F1 | OOM | OOM | $0.7201 \pm 0.0013$ | $0.6002 \pm 0.0738$ | $\mathbf{0.7219 \pm 0.0046}$ | $0.7214 \pm 0.0014$ | $0.7033 \pm 0.0294$ |
| **Amazon movies** | Precision | $0.5923 \pm 0.0046$ | OOM | $0.5878 \pm 0.0037$ | $0.5931 \pm 0.0021$ | $0.5861 \pm 0.0017$ | $0.5934 \pm 0.0010$ | $\mathbf{0.5943 \pm 0.0028}$ |
| | Recall | $0.6713 \pm 0.0038$ | OOM | $0.6711 \pm 0.0043$ | $0.6731 \pm 0.0012$ | $0.6692 \pm 0.0051$ | $0.6720 \pm 0.0084$ | $\mathbf{0.6737 \pm 0.0058}$ |
| | F1 | $0.6042 \pm 0.0072$ | OOM | $0.5917 \pm 0.0051$ | $0.5965 \pm 0.0018$ | $0.5814 \pm 0.0037$ | $0.5991 \pm 0.0064$ | $\mathbf{0.6050 \pm 0.0084}$ |
| **Yelp** | Precision | OOM | OOM | $0.6186 \pm 0.0018$ | $0.6162 \pm 0.0017$ | $0.6255 \pm 0.0026$ | $0.6295 \pm 0.0085$ | $\mathbf{0.6407 \pm 0.0016}$ |
| | Recall | OOM | OOM | $0.6686 \pm 0.0072$ | $0.6654 \pm 0.0038$ | $0.6735 \pm 0.0038$ | $0.6736 \pm 0.0108$ | $\mathbf{0.6812 \pm 0.0072}$ |
| | F1 | OOM | OOM | $0.6219 \pm 0.0103$ | $0.6085 \pm 0.0024$ | $0.6247 \pm 0.0084$ | $0.6234 \pm 0.0174$ | $\mathbf{0.6359 \pm 0.0057}$ |

the corresponding text. The detailed description of prompts can be found in Appendix C.3 and the experimental results of using different history length can be found in Appendix C.4.4. During inference, we set the temperature as 0.7 and the nucleus sampling size (*i.e., top_p*) is set as 0.95. These two hyperparameters are used to control the randomness of LLMs' output. The repetition penalty is set as 1.15 to discourage the repetitive and redundant output. The maximum number of tokens that the LLM generates is set as 1024. During fine-tuning, the LLMs are loaded in 8-bits and the rank of LoRA is set as 8. We set the batch size as 2 with only one epoch and the gradient accumulation step is set as 8. The learning rate is set as 0.0002 and we use AdamW [63] for optimization. The scaling hyperparameter $lora\_alpha$ is set as 32 and the dropout rate during fine-tuning is set as 0.05.

## 5.1 Edge Classification

Following previous work [8], we use a multi-layer perceptron to take the concatenated representations of two nodes as inputs and return the probabilities of the edge categories. The performance of different models with Bert initialization is shown in Table 3, where the best results for each dataset are shown in bold. We observe that existing models fail to achieve satisfactory performance in this task, especially on datasets with a large number of categories (*e.g.,* GDELT and ICEWS1819). This can be attributed to the fact that these models typically neglect edge information modeling in their architectures, which is extremely important for edge classification applications on DyTAGs. In Figure 4, we report the model performance with and without text attributes. It demonstrates that text information consistently helps models achieve better performance on each dataset, verifying the necessity of integrating text attributes into temporal graph modeling. Using Bert-encoded embedding as initialization serves as a preliminary strategy for dynamic textual modeling, lifting the future opportunities for more advanced embedding. We provide the complete results for other datasets in Appendix C.4.1.

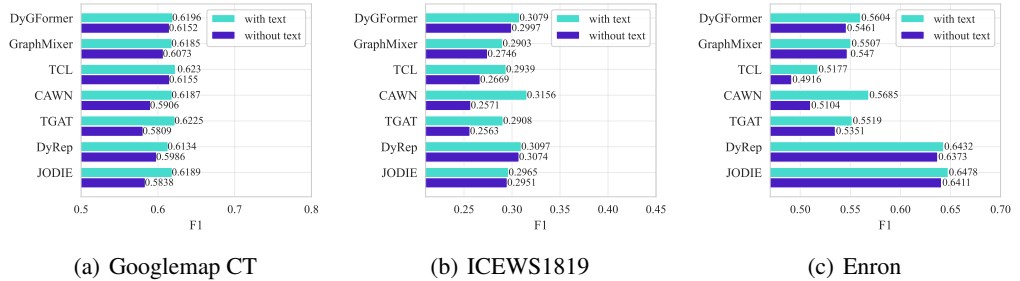

| (a) Googlemap CT | (b) ICEWS1819 | (c) Enron |

Figure 4: Edge classification performance with and without text attributes.

Table 4: AUC-ROC for future link prediction. *tr.* means transductive setting and *in.* means inductive setting. **Text** means whether to use Bert-encoded embeddings for initialization.

| | Datasets | Text | JODIE | DyRep | TGAT | CAWN | TCL | GraphMixer | DyGFormer |
|---|---|---|---|---|---|---|---|---|---|
| *tr.* | Enron | ✗ | **0.9712 ± 0.0097** | 0.9545 ± 0.0023 | 0.9511 ± 0.0011 | 0.9652 ± 0.0012 | 0.9604 ± 0.0079 | 0.9254 ± 0.0046 | 0.9653 ± 0.0015 |
| | | ✓ | 0.9731 ± 0.0052 | 0.9274 ± 0.0026 | 0.9681 ± 0.0026 | 0.9740 ± 0.0007 | 0.9618 ± 0.0025 | 0.9567 ± 0.0013 | **0.9779 ± 0.0014** |
| | ICEWS1819 | ✗ | 0.9821 ± 0.0095 | 0.9799 ± 0.0039 | 0.9787 ± 0.0065 | 0.9815 ± 0.0041 | 0.9842 ± 0.0036 | 0.9399 ± 0.0079 | **0.9865 ± 0.0024** |
| | | ✓ | 0.9741 ± 0.0113 | 0.9632 ± 0.0027 | 0.9904 ± 0.0039 | 0.9857 ± 0.0018 | **0.9923 ± 0.0012** | 0.9863 ± 0.0024 | 0.9888 ± 0.0015 |
| | Googlemap CT | ✗ | OOM | OOM | 0.8537 ± 0.0153 | **0.8543 ± 0.0027** | 0.7740 ± 0.0013 | 0.7087 ± 0.0088 | 0.7864 ± 0.0047 |
| | | ✓ | OOM | OOM | **0.9049 ± 0.0071** | 0.8687 ± 0.0063 | 0.8348 ± 0.0094 | 0.8095 ± 0.0014 | 0.8207 ± 0.0018 |
| | GDELT | ✗ | 0.9562 ± 0.0027 | 0.9477 ± 0.0011 | 0.9341 ± 0.0046 | 0.9419 ± 0.0026 | 0.9571 ± 0.0007 | 0.9316 ± 0.0021 | **0.9648 ± 0.0007** |
| | | ✓ | 0.9533 ± 0.0020 | 0.9453 ± 0.0018 | 0.9595 ± 0.0033 | 0.9600 ± 0.0061 | 0.9619 ± 0.0008 | 0.9552 ± 0.0018 | **0.9662 ± 0.0003** |
| *in.* | Enron | ✗ | 0.8745 ± 0.0041 | 0.8560 ± 0.0124 | 0.8079 ± 0.0047 | 0.8710 ± 0.0030 | 0.8363 ± 0.0068 | 0.7510 ± 0.0071 | **0.8991 ± 0.0012** |
| | | ✓ | 0.8732 ± 0.0037 | 0.7901 ± 0.0047 | 0.8650 ± 0.0032 | 0.9091 ± 0.0014 | 0.8512 ± 0.0062 | 0.8347 ± 0.0039 | **0.9316 ± 0.0015** |
| | ICEWS1819 | ✗ | 0.9115 ± 0.0081 | 0.9390 ± 0.0054 | 0.9151 ± 0.0061 | 0.9330 ± 0.0076 | 0.9471 ± 0.0011 | 0.8858 ± 0.0089 | **0.9613 ± 0.0010** |
| | | ✓ | 0.9285 ± 0.0065 | 0.9030 ± 0.0097 | 0.9706 ± 0.0054 | 0.9774 ± 0.0039 | **0.9778 ± 0.0012** | 0.9605 ± 0.0025 | 0.9630 ± 0.0027 |
| | Googlemap CT | ✗ | OOM | OOM | 0.7958 ± 0.0012 | **0.7968 ± 0.0007** | 0.7104 ± 0.0015 | 0.6675 ± 0.0033 | 0.7148 ± 0.0024 |
| | | ✓ | OOM | OOM | **0.8791 ± 0.0028** | 0.7058 ± 0.0047 | 0.7895 ± 0.0046 | 0.7543 ± 0.0018 | 0.7648 ± 0.0052 |
| | GDELT | ✗ | 0.8977 ± 0.0035 | 0.8791 ± 0.0002 | 0.7501 ± 0.0074 | 0.7909 ± 0.0010 | 0.8544 ± 0.0045 | 0.7361 ± 0.0058 | **0.9135 ± 0.0024** |
| | | ✓ | 0.8921 ± 0.0065 | 0.8917 ± 0.0007 | 0.9012 ± 0.0011 | 0.8899 ± 0.0082 | 0.9099 ± 0.0022 | 0.8942 ± 0.0035 | **0.9206 ± 0.0003** |

## 5.2 Future Link Prediction

We report the performance of different models in the AUC-ROC metric for future link prediction under transductive and inductive settings in Table 4. We have two main observations: (1) Most models achieve better performance when using text attributes. However, the performance of memory-based models (*i.e.,* JODIE and DyRep) may be degraded. This decline occurs because these models incrementally update the nodes' representations based on the numerical attributes of edges. The Bert-encoded initialization of edges will potentially mislead the update process. This observation demonstrates the limitations of simply using pre-trained embeddings to integrate the text information, showing the necessity of proposing an advanced integration strategy that can adapt to different models. (2) Larger performance improvements are observed in the inductive setting. This is because the text attributes can provide valuable information for new nodes that are difficult to distinguish using existing methods. This observation shows the effectiveness of integrating text information to handle zero-shot dynamic graph problems (*e.g.,* cold-start in recommendation [64]). We provide the complete experimental results of future link prediction in Appendix C.4.2.

## 5.3 Destination Node Retrieval

Table 5 shows the performance of the node retrieval task. For each test example, 1000 nodes including the ground truth node are randomly sampled to create the candidate set for ranking. We observe that existing models fail to achieve satisfactory performance, showing their weakness in accurately capturing the dynamic interaction preferences of nodes. Text attributes improve the model performance both in the transductive and inductive settings. This is because the descriptions and historical interactions with text can reflect the preference of a node (*e.g.,* reviews from a user reflect personalized opinion and preference). To further investigate the performance of existing models, we sample 1000 nodes that are the most recently interacted as the candidate set for each test node, which presents a more challenging setting (denoted as *historical sampling*). As shown in Figure 5, existing models perform significantly worse in the historical sampling setting, since these models largely rely on capturing structural and temporal co-occurrences, but ignore the semantic relevance

Table 5: Hits@10 for node retrieval. *tr.* means transductive setting and *in.* means inductive setting. **Text** means whether to use Bert-encoded embeddings for initialization.

| | Datasets | Text | JODIE | DyRep | TGAT | CAWN | TCL | GraphMixer | DyGFormer |
|---|---|---|---|---|---|---|---|---|---|
| *tr.* | GDELT | ✗ | **0.8733 ± 0.0095** | 0.8427 ± 0.0031 | 0.7780 ± 0.0047 | 0.7057 ± 0.0086 | 0.8134 ± 0.0079 | 0.7798 ± 0.0045 | 0.8468 ± 0.0021 |
| | | ✓ | 0.8675 ± 0.0101 | 0.8399 ± 0.0037 | 0.8817 ± 0.0035 | 0.8747 ± 0.0074 | 0.8875 ± 0.0030 | 0.8678 ± 0.0076 | **0.9016 ± 0.0011** |
| | Googlemap CT | ✗ | OOM | OOM | **0.6539 ± 0.0047** | 0.5158 ± 0.0080 | 0.4360 ± 0.0063 | 0.4076 ± 0.0017 | 0.4453 ± 0.0009 |
| | | ✓ | OOM | OOM | **0.6972 ± 0.0022** | 0.5239 ± 0.0063 | 0.5373 ± 0.0057 | 0.4855 ± 0.0028 | 0.4913± 0.0023 |
| | Amazon movies | ✗ | OOM | OOM | 0.6430 ± 0.0096 | 0.5835 ± 0.0071 | 0.6446 ± 0.0062 | 0.6478 ± 0.0035 | **0.5055 ± 0.0084** |
| | | ✓ | OOM | OOM | 0.7245 ± 0.0138 | 0.6757 ± 0.0084 | 0.7174 ± 0.0024 | 0.6805 ± 0.0026 | **0.7221 ± 0.0012** |
| | Yelp | ✗ | OOM | OOM | **0.5961 ± 0.0085** | 0.5410 ± 0.0105 | 0.5930 ± 0.0254 | 0.5745 ± 0.0248 | 0.4944 ± 0.0068 |
| | | ✓ | OOM | OOM | **0.8069 ± 0.0136** | 0.7648 ± 0.0088 | 0.5745 ± 0.0197 | 0.6585 ± 0.0064 | 0.7600 ± 0.0036 |
| *in.* | GDELT | ✗ | **0.7393 ± 0.0079** | 0.7285 ± 0.0027 | 0.6453 ± 0.0085 | 0.5752 ± 0.0038 | 0.6157 ± 0.0097 | 0.6200 ± 0.0065 | 0.7086 ± 0.0056 |
| | | ✓ | 0.7344 ± 0.0064 | 0.7266 ± 0.0061 | 0.7329 ± 0.0029 | 0.6925 ± 0.0019 | 0.7527 ± 0.0011 | 0.7123 ± 0.0046 | **0.7844 ± 0.0018** |
| | Googlemap CT | ✗ | OOM | OOM | 0.0993 ± 0.0025 | 0.0930 ± 0.0018 | 0.0997 ± 0.0021 | **0.1200 ± 0.0002** | 0.0620 ± 0.0019 |
| | | ✓ | OOM | OOM | **0.3705 ± 0.0023** | 0.2236 ± 0.0076 | 0.1769 ± 0.0009 | 0.1307 ± 0.0011 | 0.1397 ± 0.0012 |
| | Amazon movies | ✗ | OOM | OOM | 0.5559 ± 0.0125 | 0.4716 ± 0.0102 | **0.5559 ± 0.0051** | 0.5548 ± 0.0039 | 0.4617 ± 0.0138 |
| | | ✓ | OOM | OOM | **0.6461 ± 0.0093** | 0.5818 ± 0.0073 | 0.6382 ± 0.0031 | 0.5915 ± 0.0024 | 0.6481 ± 0.0016 |
| | Yelp | ✗ | OOM | OOM | **0.5282 ± 0.0160** | 0.4624 ± 0.0138 | 0.5245 ± 0.0170 | 0.4832 ± 0.0297 | 0.4265 ± 0.0103 |
| | | ✓ | OOM | OOM | **0.7204 ± 0.0130** | 0.6777 ± 0.0104 | 0.4832 ± 0.0134 | 0.5718 ± 0.0084 | 0.6825 ± 0.0036 |

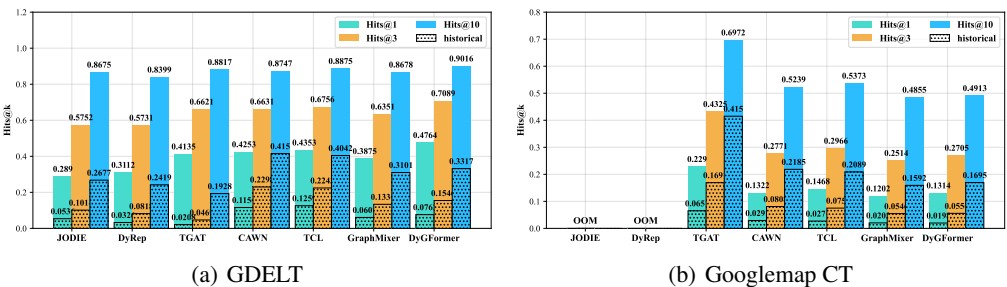

(a) GDELT  (b) Googlemap CT

Figure 5: Node retrieval performance using random sampling and historical sampling.

Table 6: Precision, Recall and F1 of BERTscore of different LLMs for textural relation generation. The number of test samples is 500 per dataset.

| | Googlemap CT | | | Amazon movies | | | Stack elec | | |
|---|---|---|---|---|---|---|---|---|---|
| | Precision | Recall | F1 | Precision | Recall | F1 | Precision | Recall | F1 |
| GPT 3.5 turbo | 79.89 | **84.13** | 81.91 | 79.79 | 83.61 | 81.63 | 80.52 | 81.96 | 81.21 |
| GPT 4o | 78.33 | 84.06 | 81.07 | 78.68 | **84.20** | 81.33 | 78.30 | 82.37 | 80.26 |
| Llama3-8b | 78.62 | 83.84 | 81.12 | 78.48 | 83.97 | 81.09 | 79.91 | 82.35 | 81.09 |
| Mistral-7b | **80.21** | 84.05 | **82.07** | 79.81 | 84.05 | **81.84** | 80.25 | **82.61** | 81.40 |
| Vicuna-7b | 80.04 | 83.79 | 81.85 | **80.23** | 83.60 | 81.83 | **80.65** | 82.37 | **81.46** |
| Vicuna-13b | 80.14 | 84.00 | 81.99 | 77.59 | 83.56 | 80.39 | 80.57 | 82.20 | 81.33 |

and long-range dependencies that are important in real-world applications. More experimental results are provided in Appendix C.4.3.

## 5.4 Textural Relation Generation

Generating the textual content of future interactions within certain node pairs remains an under-explored challenge, which requires a language model to understand the dynamics and textural description within a graph structure. We evaluate six LLMs on three datasets derived from different real-world scenarios. For instance, on the `Googlemap CT` dataset, the input sentence is constructed by the user's historical reviews and textural description of the destination. Then, LLM is prompted to generate potential reviews from the user to the future destination. Instead, the LLM is prompted to generate reviews for target

Table 7: Performance of LLMs after SFT for the relation generation task in terms of BERTscore (F1).

| | Googlemap CT | Stack elec |
|---|---|---|
| Llama3-8b | 81.12 | 81.09 |
| Llama3-8b + SFT | 81.84 (0.72↑) | 81.97 (0.88↑) |
| Vicuna-7b | 81.85 | 81.46 |
| Vicuna-7b + SFT | **85.67** (3.82↑) | 82.67 (1.21↑) |
| Vicuna-13b | 81.99 | 81.33 |
| Vicuna-13b + SFT | 84.67 (2.68↑) | **82.73** (1.40↑) |

movies and answers to target questions on the `Amazon movies` and `Stack elec` datasets, respectively. See Appendix C.3 for dataset-specific prompts. As shown in Table 6, we observe that

open-source LLMs such as Mistral and Vicuna perform comparably well to proprietary LLMs in this task. To further improve their performance on relation generation, we extract 10,000 node pair interactions associated with textural descriptions from the datasets for LLM supervised fine-tuning (SFT). Results in Table 7 demonstrate that LLMs consistently achieve enhanced performance in textural generation after supervised fine-tuning. Especially, Vicuna-7b benefits the most from SFT. We provide the ablation study on different information provided to LLM in Appendix C.4.4.

## 6   Discussion

**Limitations and Future Directions**. While the proposed DTGB represents a significant advancement in the study of DyTAGs, there are areas ripe for further exploration. Our extensive benchmark experiments reveal that current dynamic graph learning algorithms and large language models (LLMs) exhibit varying degrees of effectiveness when handling the complex interactions between the dynamic graph structure and textural attributes. This finding highlights the potential for even greater improvements and innovations in this field in the future.

A particularly exciting future direction is the design of temporal graph tokens that can directly incorporate dynamic graph information into LLMs for reasoning and dynamics-aware generation. By designing representations that seamlessly blend structural and temporal aspects of the graphs with their text attributes, these tokens could potentially enhance the ability of LLMs to capture and utilize the dynamic nature of DyTAGs. This approach promises to improve performance in a range of applications, such as real-time recommendation systems, dynamic knowledge graphs, and evolving social network analysis.

Another notable challenge is the scalability issue when handling large-scale DyTAGs, especially given the potentially long text descriptions associated with nodes and edges. The complexity of encoding long sequences and integrating them with dynamic graph structures can lead to computational overhead. Addressing this scalability issue is a crucial future direction to ensure that models can efficiently process large-scale graphs with extensive textual attributes, paving the way for more practical and robust applications in real-world scenarios.

**Broader Impact**. The broader impact of DTGB lies in its ability to drive advancements in dynamic text-attributed graph research by providing a comprehensive benchmark for evaluating models. The broader impact can extend to numerous societal and technological domains, such as social media and real-time recommendation systems. Furthermore, advancements driven by DTGB that can integrate dynamic graph learning with natural language processing could lead to methodological enhancement in fields such as healthcare, finance, and cybersecurity, where understanding the evolving relationships and information is critical for decision-making and risk management. Overall, DTGB has the potential to drive significant improvements in how complex, dynamic data is harnessed and utilized across various sectors.

## 7   Conclusion

We propose the first comprehensive benchmark DTGB specifically for dynamic text-attributed graphs (DyTAGs). We collect and provide eight carefully processed dynamic text-attributed graph datasets from diverse domains. Based on these datasets, we comprehensively investigate the performance of existing dynamic graph learning models and large language models (LLMs) in four real-world-driven tasks. Our experimental results validate the utility of DTGB and provide insights for further technical advancements. The limitation of this work is that we did not incorporate the high-order graph context in the textual relation generation task, due to the maximum input length of LLMs. Therefore, in the future, we will investigate how to efficiently use LLM to handle high-order dynamic topology and long-range evolving texts within DyTAGs.

## Acknowledgments and Disclosure of Funding

This work is supported by the Shenzhen Science and Technology Program (No. JCYJ20210324121213037) and Guangxi Key Research and Development Program (No. Guike AB24010112).

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

# A Datasets

## A.1 Dataset Description

**Dataset Format**. For each dataset in DTGB, we provide three different files. We store each edge as a tuple in the edge_list.csv file, which includes id of the source node, id of the target node, id of the relation between them, the occurring timestamp, and the edge category. We use entity_text.csv and relation_text.csv to store the text attributes of nodes and edges for each dataset. Each file includes the mapping from node and relation ids to the corresponding raw text descriptions. All the datasets and codes to reproduce the results in this paper are available at `https://github.com/zjs123/DTGB`. We provide a detailed description and the link to the raw resources of each dataset as follows.

**Enron**[3]. This dataset is derived from the email communications between employees of the ENRON energy corporation over three years (1999-2002). The nodes indicate the employees while the edges are e-mails among them. The text attribute of each node is extracted from the department and position of the employee (if available). The text attribute of each edge is the raw text of e-mails. Non-English statements, abnormal symbols, and tables are removed from the raw text and we perform length truncation on these e-mails. The edge categories are extracted from the e-mail archive of the raw resource. There are 10 kinds of categories such as *calendar*, *notes*, and *deal communication*. We order edges in this dataset based on the sending timestamps of e-mails.

**GDELT**[4]. This dataset is derived from the Global Database of Events, Language, and Tone project, which is an initiative to construct a catalog of political behavior across all countries of the world. Nodes in this dataset indicate political entities such as *United States* and *Kim Jong UN*. We directly use the names of these entities as their textual attributes. Edges in this dataset represent the relationships between entities (e.g., *President of* and *Make Statement*). We use the descriptions of these relationships as the textual attributes of edges. Each edge category refers to a kind of political relationship or behavior. We order edges in this dataset based on the occurring timestamps of these political events.

**ICEWS1819**[5]. This dataset is derived from the Integrated Crisis Early Warning System project, which is also a temporal knowledge graph for political events. We extract events from 2018-01-01 to 2019-12-31 to construct this dataset. We organize the name, sector, and nationality of each political entity as its text attribute, while the edge text attributes are the descriptions of the political relationships. The edge categories refer to the types of political relationships or behavior. Similar to the GDELT dataset, we order edges in this dataset based on the occurring timestamps of the political events. Note that the major difference between the ICEWS1819 and GDELT datasets is that first, the time granularity of GDELT is 15 minutes, while that of ICEWS1819 is 24 hours. Therefore GDELT describes political interactions in a more fine-grained way. Second, ICEWS1819 has a 4 times larger node set compared with GDELT and thus represents a more sparse scenario.

**Stack elec**[6]. Stack Exchange Data is an anonymized dump of all user-contributed content on various stack exchange sites. It includes questions, answers, comments, tags, and other related data from these sites. We regard the questions and users in these sites as nodes, while the answers and comments from users to questions are regarded as text-attributed edges, subsequently constructing a dynamic bipartite graph that describes the multi-round dialogue between users and questions. We extract all the questions related to electronic techniques as well as the corresponding answers and comments to construct the Stack elec dataset. For user nodes, we use the self-introductions of users as their text attributes, which describe the technical areas that the user is familiar with. For the question node, we use the title and the body of each question post as its text attribute. We use the raw text of answers and comments as the text attributes of edges. We construct two categories based on the voting of each answer: *Useful* if the voting count is larger than 1, otherwise *Useless*. We order edges in this dataset based on the answering timestamps from users.

---

[3] `https://www.cs.cmu.edu/~enron/`
[4] `https://www.gdeltproject.org/`
[5] `https://dataverse.harvard.edu/dataverse/icews`
[6] `https://archive.org/details/stackexchange`

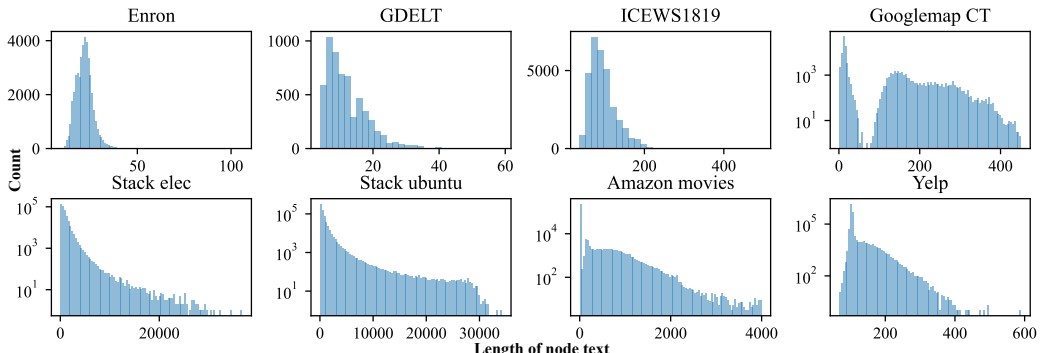

Figure 6: Distribution of node text length on DTGB datasets.

**Stack ubuntu**[7]. This is another dataset from Stack Exchange Data, which contains all the questions related to the Ubuntu system. Besides the size and the topic, the biggest difference between this dataset and `Stack elec` is that the answers in this dataset are usually a mixture of codes and natural language, which brings more challenges to the understanding of the semantic context of interactions.

**Googlemap CT**[8]. This dataset is extracted from the Google Local Data project, which contains review information on Google map as well as the user and business information up to September 2021 in the United States. We extract all the business entities from Connecticut State to construct this dataset. Nodes are users and business entities while edges are reviews from users to businesses. Only the business nodes are enriched with text attributes, containing the name, address, category, and self-introduction of the business entity. The edge text attributes are the raw text of user reviews. The edge categories are integers from 1 to 5, derived from the ratings from users to businesses. We have removed emojis and meaningless characters from reviews. Edges in this dataset are ordered based on the review timestamps from users.

**Amazon movies**[9]. This dataset is extracted from the Amazon Review Data project, which contains product reviews and metadata from Amazon spanning May 1996 to July 2014. To construct this dataset, we extract products in the class of *Movies and TV* and the corresponding reviews. The text attribute of each product node contains its name, category, description, and rank score. The text attributes of edges are review text from users to products. Similarly, the edge categories are integers from 1 to 5, derived from the ratings from users to businesses. We still order edges in this dataset based on the review timestamps from users.

**Yelp**[10]. This dataset is extracted from the Yelp Open Dataset project which contains reviews of restaurants, shopping centers, hotels, tourism, and other businesses from users. The text attribute of each business node contains its name, address, city, and category. The text attribute of each user node contains its first name, number of reviews, and register time. Edge text is the reviews from users to businesses. Edge categories are also the ratings from 1 to 5. All the edges are ordered based on the review timestamps from users.

### A.2 Dataset Analysis

Here, we provide more statistical analysis for datasets in the DTGB benchmark. As illustrated in Figure 6, we can see that datasets `Stack elec` and `Stack ubuntu` have significantly longer node text than other datasets. This is because question text in these datasets usually contains references and the introduction of background, which will bring unique challenges in understanding the node semantics. One interesting observation is that the distribution of the `Googlemap CT` dataset is bimodal. This is because some businesses in this dataset lack *description* raw text, and thus have significantly shorter text length than others. In Figure 7, we can see that most datasets in the DTGB benchmark show long-tail distribution on node degrees, which meets the real world. In Figure 8, we

---

[7]https://archive.org/details/stackexchange

[8]https://datarepo.eng.ucsd.edu/mcauley_group/gdrive/googlelocal/

[9]https://cseweb.ucsd.edu/~jmcauley/datasets/amazon_v2/

[10]https://www.yelp.com/dataset

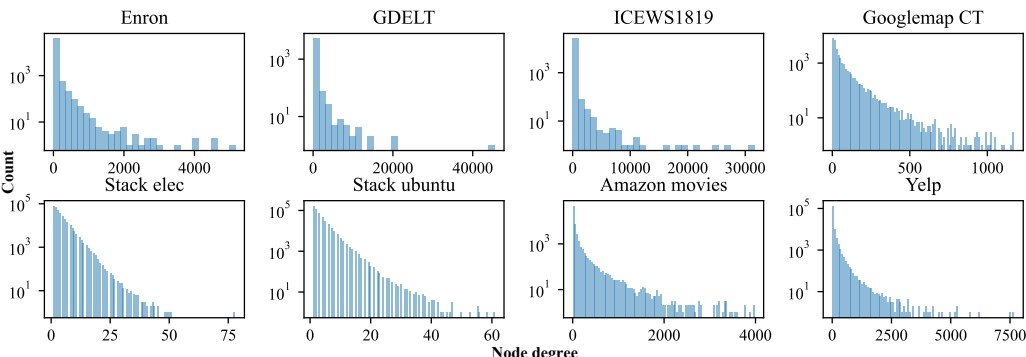

Figure 7: Distribution of node degree on DTGB datasets.

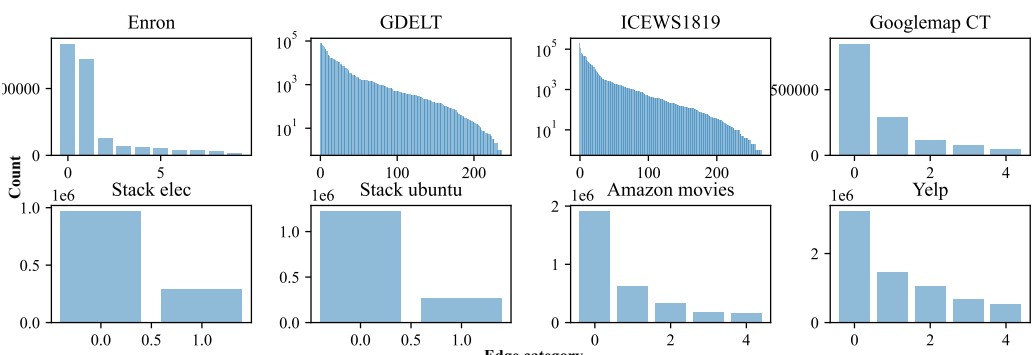

Figure 8: Distribution of the number of edges for each category on DTGB datasets.

can see that the categories in these datasets are non-uniformly distributed. We preserve such skewed distribution to faithfully reflect the challenges in real-world applications and leave opportunities to handle these challenges through the incorporation of dynamic graph structure and text semantic modeling.

## A.3 License

All the used codes and datasets are publicly available and permit usage for research purposes under either MIT License or Apache License 2.0.

## B Notations

In Table 8, we summarize important notations used in this paper and provide the corresponding descriptions.

## C Experiments

### C.1 Baselines

#### C.1.1 Temporal Graph Models

**JODIE [54].** This model learns to project/forecast the embedding trajectories into the future to make predictions about the entities and their interactions. Two coupled recurrent neural networks are used to update the states of entities and a projection operation is used to get the future representation trajectory of each entity.

**DyRep [42].** This model proposes a recurrent architecture to update node states upon each interaction. It uses a deep temporal point process model to capture the dynamics of the observed processes. This

Table 8: Important notations and descriptions.

| Notation | Description |
|---|---|
| $\mathcal{G}$ | A dynamic text-attributed graph. |
| $\mathcal{V}, \mathcal{E}$ | Node and edge sets of $\mathcal{G}$. |
| $\mathcal{G}_T$ | Subgraph of $\mathcal{G}$ which contains nodes and edges appeared before timestamp $T$. |
| $\mathcal{V}_T, \mathcal{E}_T$ | Node and edge sets of $\mathcal{G}_T$. |
| $\mathcal{D}$ | Set of node text descriptions. |
| $\mathcal{R}$ | Set of edge text descriptions. |
| $\mathcal{L}$ | Set of edge categories. |
| $\mathcal{T}$ | Set of observed timestamps. |
| $u, v$ | Nodes in the dynamic text-attributed graph. |
| $(u, v)$ | An edge in the dynamic text-attributed graph which connects nodes $u$ and $v$. |
| $d_v$ | Text description of node $v$. |
| $r_{u,v}$ | Text description of edge $(u, v)$. |
| $l_{u,v}$ | Category of edge $(u, v)$. |
| $t_{u,v}$ | Occurring timestamp of edge $(u, v)$. |

model is further parameterized by a temporal-attentive network that encodes temporally evolving structural information into node representations which in turn drives the nonlinear evolution of the observed graph dynamics.

**TGAT [37].** This model aims to efficiently aggregate temporal-topological neighborhood features as well as to learn the time-feature interactions. It uses the self-attention mechanism as a building block. A functional time encoding technique based on the classical Bochner's theorem from harmonic analysis is used to capture temporal patterns.

**CAWN [39].** This model adopts an anonymization strategy based on a set of sampled walks to explore the causality of network dynamics and generate inductive node identities. Then, a neural network model is used to encode these sampled walks and aggregate them to obtain the final node representation.

**TCL [41].** This model designs a two-stream encoder that separately processes temporal neighborhoods associated with the two target interaction nodes. Then, a graph-topology-aware Transformer is proposed to consider both graph topology and temporal information to learn node representations. Cross-attention operation is also incorporated to learn the relevance between two interaction nodes.

**GraphMixer [40].** This model shows the effectiveness of the fixed-time encoding function in modeling dynamic interactions. The proposed simple architecture consists of three components: a link encoder used to summarize the information from temporal links, a node encoder used to summarize node information, and a link classifier that performs link prediction.

**DyGFormer [8].** This model learns node representations from nodes' historical first-hop interactions. It uses a neighbor co-occurrence encoding scheme to explore the correlations between nodes based on their historical sequences. A patching technique is also proposed to divide each sequence into multiple patches, which allows the model to effectively benefit from longer histories.

### C.1.2 Large Language Models

**Mistral (7B) [55].** This language model leverages grouped-query attention (GQA) for faster inference, coupled with sliding window attention (SWA) to effectively handle sequences of arbitrary length with a reduced inference cost. We use the Mistral-7B-Instruct-v0.2 version in our experiment.

**Llama-3 (8B) [57].** This is an auto-regressive language model that uses an optimized Transformer architecture. The tuned versions use supervised fine-tuning (SFT) and reinforcement learning with human feedback (RLHF) to align with human preferences for helpfulness and safety. We use the 8-billion-parameter instruction version of Llama-3 in our experiment.

**Vicuna (7B/13B) [65].** Vicuna is fine-tuned from Llama 2 with supervised instruction fine-tuning. The training data is from the user-shared conversations collected on the internet. We use the vicuna-7b-v1.5 version and vicuna-13b-v1.5 version in our experiment.

**GPT3.5-turbo and GPT4o**[11]. Generative pre-trained Transformer 3.5 (GPT-3.5) is a sub-class of GPT-3 models created by OpenAI in 2022. GPT-4o was released on 13 May 2024, which achieves state-of-the-art results in voice, multilingual, and vision benchmarks.

## C.2 Metrics

In this section, we describe the evaluation metrics used in four benchmark tasks. For the edge classification task, we use the weighted average of different categories to avoid the influence of imbalance category distribution. For future link prediction, we follow the previous work [8] to use the Average Precision and AUC-ROC metrics. For the destination node retrieval task, we employ the widely used Hits@k metric. For the textual relation generation task, we use the Bertscore to concentrate on semantic relevance between ground truth and the generated sentence.

### C.2.1 Edge Classification

**Weighted Precision**. This metric is the weighted average of the precisions of different classes. The weights are typically the number of instances for each class or other meaningful metrics.

$$\text{Weighted Precision} = \frac{\sum_{i=1}^{n} w_i \cdot P_i}{\sum_{i=1}^{n} w_i}, \tag{1}$$

where $w_i$ is the weight of the $i$-th class, and $P_i$ is the precision of the $i$-th class.

**Weighted Recall**. This metric is the weighted average of the recalls of different classes, similar to Weighted Precision.

$$\text{Weighted Recall} = \frac{\sum_{i=1}^{n} w_i \cdot R_i}{\sum_{i=1}^{n} w_i}, \tag{2}$$

where $w_i$ is the weight of the $i$-th class, and $R_i$ is the recall of the $i$-th class.

**Weighted F1-score**. This metric is the weighted average of the F1-scores of different classes, combining the benefits of Precision and Recall.

$$\text{Weighted F1-score} = \frac{\sum_{i=1}^{n} w_i \cdot F1_i}{\sum_{i=1}^{n} w_i}, \tag{3}$$

where $w_i$ is the weight of the $i$-th class, and $F1_i$ is the F1-score of the $i$-th class, calculated as:

$$F1_i = 2 \cdot \frac{P_i \cdot R_i}{P_i + R_i}. \tag{4}$$

### C.2.2 Future Link Prediction

**Average Precision**. This metric is the average of precision values at different recall levels. It is the area under the Precision-Recall curve.

$$\text{Average Precision} = \sum_{k=1}^{n} (R_k - R_{k-1}) P_k, \tag{5}$$

where $R_k$ is the recall at the $k$-th threshold, and $P_k$ is the precision at the $k$-th threshold.

**AUC-ROC**. AUC-ROC (Area Under the Receiver Operating Characteristic Curve) is a performance measurement for classification problems at various threshold settings. The ROC is a probability curve, and AUC represents the degree or measure of separability.

$$\text{AUC-ROC} = \int_0^1 \text{TPR}(\text{FPR}) \, d(\text{FPR}), \tag{6}$$

where TPR is the true positive rate, and FPR is the false positive rate. The ROC curve is created by plotting the TPR against the FPR at various threshold settings.

---

[11]https://openai.com/

### C.2.3 Destination Node Retrieval

**Hits@k**. Hits@k (Hits at $k$) is a metric used in information retrieval and recommendation systems to evaluate the effectiveness of a model in retrieving relevant items. It measures the proportion of times the true positive item is found within the top-$k$ predictions.

$$\text{Hits@k} = \sum_i \frac{\mathbb{I}\left(\text{rank}_i \leq k\right)}{Q}, \tag{7}$$

where $\mathbb{I}(*)$ is the indicator function (returns 1 if the condition is true, otherwise 0), and $Q$ represents the total number of test samples.

### C.2.4 Textual Relation Generation

**Bertscore**. This metric evaluates the semantic similarity between candidate and reference texts using the BERT model.

$$\text{BERTscore}(c, r) = \frac{1}{|c|} \sum_{i=1}^{|c|} \max_{j=1,\ldots,|r|} \text{BERT}(c_i, r_j), \tag{8}$$

where $c$ and $r$ are the candidate and reference texts, respectively, and $\text{BERT}(c_i, r_j)$ is the similarity score between the $i$-th word in the candidate text and the $j$-th word in the reference text, computed by the BERT model.

**Precision**. In the context of BERTscore, precision measures the accuracy of the generated text by assessing the proportion of relevant text generated among all the text produced by the model. A high precision indicates that the generated text is highly relevant and accurate compared with the reference text. The complete score matches each token in $\hat{x}$ to a token in $x$ to compute precision. Then, we use greedy matching to maximize the matching similarity score, where each token is matched to the most similar token in the other sentence. The precision score for BERTscore is calculated as:

$$P_{\text{BERT}} = \frac{1}{|\hat{x}|} \sum_{\hat{x}_j \in \hat{x}} \max_{x_i \in x} \mathbf{x}_i^\top \hat{\mathbf{x}}_j. \tag{9}$$

**Recall**. In the context of BERTscore, recall evaluates the completeness of the generated text by measuring the proportion of relevant text captured by the model among all the relevant text present in the reference text. A high recall indicates that the model is good at capturing relevant information from the reference text in its generated output. The complete score matches each token in $x$ to a token in $\hat{x}$ to compute recall and it is calculated as:

$$R_{\text{BERT}} = \frac{1}{|x|} \sum_{x_i \in x} \max_{\hat{x}_j \in \hat{x}} \mathbf{x}_i^\top \hat{\mathbf{x}}_j. \tag{10}$$

**F1-score**. In the context of BERTscore, F1-score is the harmonic mean of precision and recall. It balances between precision and recall, providing a single metric to assess the overall performance of the model in generating relevant text. The F1-score for BERTscore is calculated as:

$$F_{\text{BERT}} = 2 \frac{P_{\text{BERT}} \cdot R_{\text{BERT}}}{P_{\text{BERT}} + R_{\text{BERT}}}. \tag{11}$$

### C.3 Prompts for Textual Relation Generation

As illustrated in Figure 9, we give an example of prompts used in the textual relation generation task, which visually illustrates the presentation of the instruction to the language model. Note that for different datasets, some keywords in the prompt will be consequently changed. For the `Stack elec` and `Stack ubuntu` datasets, the words *item* and *review* will be changed to *question* and *answer*. For the `ICEWS1819` and `GDELT` datasets, words *item* and *user* will be unified as *entity*, and the word *review* will be changed to *relation*. For the `Enron` dataset, words *item* and *user* will be unified as *user*, and the word *review* will be changed to *e-mail*.

> **Description of item A:** Name:xxx, Introduction: Family-run pizzeria with standout baby clam pies in a cozy space decorated with family photos.
>
> **Recent reviews of item A from other users:**
> 1. My favorite pizza of all time!
> 2. Totally Great Pizza get the special.
>
> **Recent reviews of user P to other items:**
> 1. item: Name: yyy, Introduction: ...
>    review: The BBQ is fabulous. I has the #1 Combo with pulled pork, brisket, and ribs. AWESOME!
> 2. item: Name: zzz, Introduction: ...
>    review: Excellent wine and spirits store.
>
> **Human Question:** If User P visit Item A in the next time, please give me three possible review of User P to Item A.
>
> **LLM Response:**
> 1. Delicious Pizza!!!
> 2. One of the few authentic apizza spots.
> 3. The sausage is their best pie, simply because it's made by them.

Figure 9: Example of prompts used for inference and fine-tuning in the textual relation generation task (a case in the `Googlemap CT` dataset).

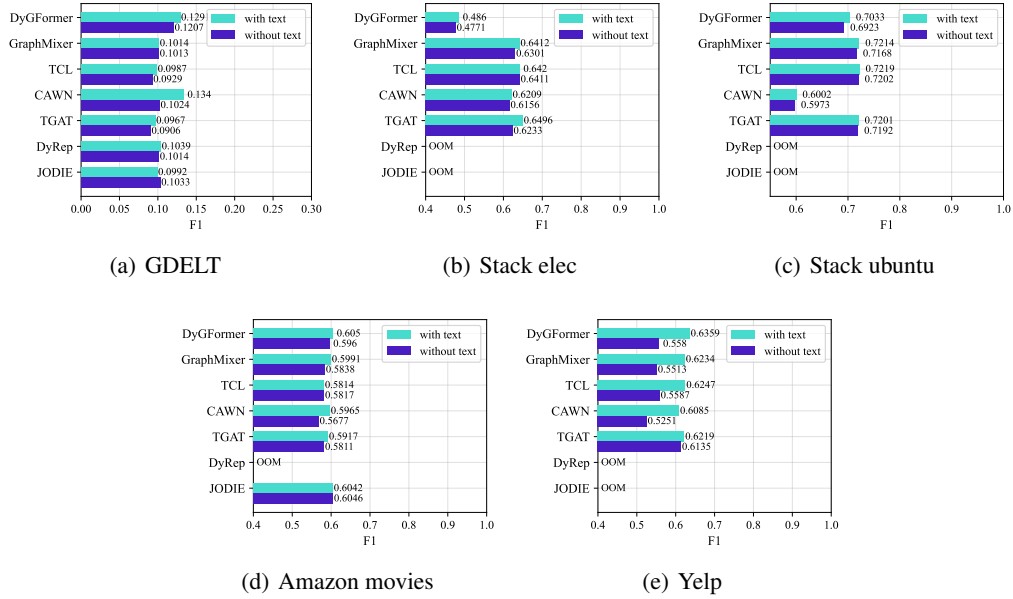

Figure 10: Edge classification performance with and without text attributes.

## C.4 Additional Experiment Results

### C.4.1 Edge Classification

As illustrated in Figure 10, we can see that text information can improve the edge classification performance of existing models on datasets from different domains, indicating the effectiveness of integrating edge text attributes in discriminating the interaction types. However, some models have performance degradation with text information, such as TCL on the `Amazon movies` dataset, showing the weakness of some models in handling rich semantic information. This observation demonstrates the necessity of designing flexible methods to introduce text information to existing dynamic graph learning methods.

Table 9: AUC-ROC for future link prediction. *tr.* means transductive setting and *in.* means inductive setting. **Text** means whether to use Bert-encoded embeddings for initialization.

| | Datasets | Text | JODIE | DyRep | TGAT | CAWN | TCL | GraphMixer | DyGFormer |
|---|---|---|---|---|---|---|---|---|---|
| *tr.* | Enron | ✗ | **0.9712 ± 0.0097** | 0.9545 ± 0.0023 | 0.9511 ± 0.0011 | 0.9652 ± 0.0012 | 0.9604 ± 0.0079 | 0.9254 ± 0.0046 | 0.9653 ± 0.0015 |
| | | ✓ | 0.9731 ± 0.0052 | 0.9274 ± 0.0026 | 0.9681 ± 0.0026 | 0.9740 ± 0.0007 | 0.9618 ± 0.0025 | 0.9567 ± 0.0013 | **0.9779 ± 0.0014** |
| | ICEWS1819 | ✗ | 0.9821 ± 0.0095 | 0.9799 ± 0.0039 | 0.9787 ± 0.0065 | 0.9815 ± 0.0041 | 0.9842 ± 0.0036 | 0.9399 ± 0.0079 | **0.9865 ± 0.0024** |
| | | ✓ | 0.9741 ± 0.0113 | 0.9632 ± 0.0027 | 0.9904 ± 0.0039 | 0.9857 ± 0.0018 | **0.9923 ± 0.0012** | 0.9863 ± 0.0024 | 0.9888 ± 0.0015 |
| | Googlemap CT | ✗ | OOM | OOM | 0.8537 ± 0.0153 | **0.8543 ± 0.0027** | 0.7740 ± 0.0013 | 0.7087 ± 0.0088 | 0.7864 ± 0.0047 |
| | | ✓ | OOM | OOM | **0.9049 ± 0.0071** | 0.8687 ± 0.0063 | 0.8348 ± 0.0094 | 0.8095 ± 0.0014 | 0.8207 ± 0.0018 |
| | GDELT | ✗ | 0.9562 ± 0.0027 | 0.9477 ± 0.0011 | 0.9341 ± 0.0046 | 0.9419 ± 0.0026 | 0.9571 ± 0.0007 | 0.9316 ± 0.0021 | **0.9648 ± 0.0007** |
| | | ✓ | 0.9533 ± 0.0020 | 0.9453 ± 0.0018 | 0.9595 ± 0.0033 | 0.9600 ± 0.0061 | 0.9619 ± 0.0008 | 0.9552 ± 0.0018 | **0.9662 ± 0.0003** |
| | Stack elec | ✗ | OOM | OOM | 0.9609 ± 0.0010 | 0.9610 ± 0.0013 | 0.8796 ± 0.0089 | 0.9604 ± 0.0006 | **0.9788 ± 0.0018** |
| | | ✓ | OOM | OOM | 0.9709 ± 0.0014 | 0.9631 ± 0.0007 | 0.9578 ± 0.0106 | 0.9673 ± 0.0011 | **0.9798 ± 0.0006** |
| | Stack ubuntu | ✗ | OOM | OOM | 0.9180 ± 0.0520 | **0.9564 ± 0.0003** | 0.9508 ± 0.0012 | 0.9522 ± 0.0006 | 0.9522 ± 0.0027 |
| | | ✓ | OOM | OOM | 0.9490 ± 0.0018 | **0.9567 ± 0.0004** | 0.9516 ± 0.0003 | 0.9494 ± 0.0028 | 0.9526 ± 0.0035 |
| | Amazon movies | ✗ | OOM | OOM | 0.8698 ± 0.0010 | 0.8561 ± 0.0003 | 0.8711 ± 0.0007 | **0.8726 ± 0.0003** | 0.8713 ± 0.0005 |
| | | ✓ | OOM | OOM | 0.9064 ± 0.0014 | 0.8936 ± 0.0012 | 0.9050 ± 0.0012 | 0.8894 ± 0.0008 | **0.9100 ± 0.0006** |
| | Yelp | ✗ | OOM | OOM | **0.8646 ± 0.0037** | 0.8445 ± 0.0021 | 0.8593 ± 0.0014 | 0.8554 ± 0.0013 | 0.8641 ± 0.0007 |
| | | ✓ | OOM | OOM | 0.9487 ± 0.0029 | 0.9349 ± 0.0026 | **0.9528 ± 0.0018** | 0.8927 ± 0.0021 | 0.9407 ± 0.0010 |
| *in.* | Enron | ✗ | 0.8745 ± 0.0041 | 0.8560 ± 0.0124 | 0.8079 ± 0.0047 | 0.8710 ± 0.0030 | 0.8363 ± 0.0068 | 0.7510 ± 0.0071 | **0.8991 ± 0.0012** |
| | | ✓ | 0.8732 ± 0.0037 | 0.7901 ± 0.0047 | 0.8650 ± 0.0032 | 0.9091 ± 0.0014 | 0.8512 ± 0.0062 | 0.8347 ± 0.0039 | **0.9316 ± 0.0015** |
| | ICEWS1819 | ✗ | 0.9115 ± 0.0081 | 0.9390 ± 0.0054 | 0.9151 ± 0.0061 | 0.9330 ± 0.0076 | 0.9471 ± 0.0011 | 0.8858 ± 0.0089 | **0.9613 ± 0.0010** |
| | | ✓ | 0.9285 ± 0.0065 | 0.9030 ± 0.0097 | 0.9706 ± 0.0054 | 0.9774 ± 0.0039 | **0.9778 ± 0.0012** | 0.9605 ± 0.0025 | 0.9630 ± 0.0027 |
| | Googlemap CT | ✗ | OOM | OOM | 0.7958 ± 0.0012 | **0.7968 ± 0.0007** | 0.7104 ± 0.0015 | 0.6675 ± 0.0033 | 0.7148 ± 0.0024 |
| | | ✓ | OOM | OOM | **0.8791 ± 0.0028** | 0.7058 ± 0.0047 | 0.7895 ± 0.0046 | 0.7543 ± 0.0018 | 0.7648 ± 0.0052 |
| | GDELT | ✗ | 0.8977 ± 0.0035 | 0.8791 ± 0.0002 | 0.7501 ± 0.0074 | 0.7909 ± 0.0010 | 0.8544 ± 0.0045 | 0.7361 ± 0.0058 | **0.9135 ± 0.0024** |
| | | ✓ | 0.8921 ± 0.0065 | 0.8917 ± 0.0007 | 0.9012 ± 0.0011 | 0.8899 ± 0.0082 | 0.9099 ± 0.0022 | 0.8942 ± 0.0035 | **0.9206 ± 0.0003** |
| | Stack elec | ✗ | OOM | OOM | 0.7800 ± 0.0024 | 0.7833 ± 0.0089 | 0.7243 ± 0.0074 | 0.7856 ± 0.0039 | **0.8578 ± 0.0035** |
| | | ✓ | OOM | OOM | 0.8423 ± 0.0018 | 0.7963 ± 0.0074 | 0.7689 ± 0.0051 | 0.8232 ± 0.0031 | **0.8607 ± 0.0015** |
| | Stack ubuntu | ✗ | OOM | OOM | 0.7213 ± 0.0549 | 0.7892 ± 0.0039 | 0.7712 ± 0.0036 | 0.7752 ± 0.0021 | **0.7895 ± 0.0018** |
| | | ✓ | OOM | OOM | 0.7655 ± 0.0019 | **0.7871 ± 0.008** | 0.7717 ± 0.0027 | 0.7870 ± 0.0032 | 0.7773 ± 0.0047 |
| | Amazon movies | ✗ | OOM | OOM | **0.8186 ± 0.0017** | 0.7910 ± 0.0012 | 0.8129 ± 0.0006 | 0.8139 ± 0.0008 | 0.8140 ± 0.0007 |
| | | ✓ | OOM | OOM | 0.8706 ± 0.0023 | 0.8492 ± 0.0008 | 0.8677 ± 0.0009 | 0.8418 ± 0.0012 | **0.8733 ± 0.0005** |
| | Yelp | ✗ | OOM | OOM | **0.8090 ± 0.0011** | 0.7843 ± 0.0018 | 0.8011 ± 0.0009 | 0.7951 ± 0.0006 | 0.8058 ± 0.0005 |
| | | ✓ | OOM | OOM | 0.9173 ± 0.0008 | 0.8995 ± 0.0005 | **0.9233 ± 0.0011** | 0.8452 ± 0.0014 | 0.9067 ± 0.0009 |

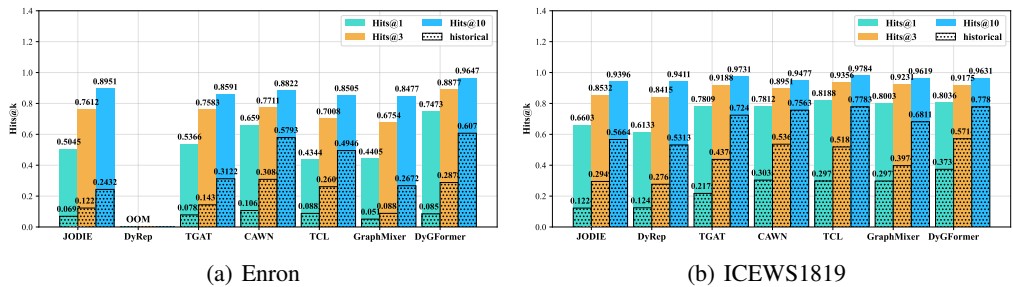

(a) Enron           (b) ICEWS1819

Figure 11: Node retrieval performance using random sampling and historical sampling.

### C.4.2 Future Link Prediction

As illustrated in Table 9 and Table 10, we report the complete results of the future link prediction task. We can see that the memory-based models (*i.e.,* JODIE and DyRep) have scalability problems facing large dynamic graphs (*e.g.,* `Stack ubuntu` and `Yelp`). Since most dynamic graphs in the real world are large and enriched with text attributes while many applications have real-time requirements (*e.g.,* anomaly detection), it requires lightweight models with accuracy and efficiency to adapt to real-world scenarios. Furthermore, we can see that even though some models get good performance in the future link prediction task, they fail to achieve satisfactory performance in the destination node retrieval task (*e.g.,* GraphMixer on the `Enron` dataset), showing the limitations of existing models.

### C.4.3 Destination Node Retrieval

As illustrated in Table 11 and Table 12, we provide more results of the destination node retrieval task in the Hits@1 and Hits@3 metrics. As illustrated in Figure 11, we provide the ablation study result of sampling strategies on the `Enron` and `ICEWS1819` datasets. The results meet our observations in Section 5.3 and indicate the limitations of existing models.

Table 10: AP for future link prediction. *tr.* means transductive setting and *in.* means inductive setting. **Text** means whether to use Bert-encoded embeddings for initialization.

| | Datasets | Text | JODIE | DyRep | TGAT | CAWN | TCL | GraphMixer | DyGFormer |
|---|---|---|---|---|---|---|---|---|---|
| *tr.* | Enron | ✗ | 0.9566 ± 0.0047 | 0.9498 ± 0.0059 | 0.9490 ± 0.0015 | 0.9664 ± 0.0009 | 0.9594 ± 0.0032 | 0.9138 ± 0.0035 | **0.9716 ± 0.0018** |
| | | ✓ | 0.9553 ± 0.0051 | 0.9066 ± 0.0076 | 0.9668 ± 0.0026 | 0.9756 ± 0.0008 | 0.9603 ± 0.0018 | 0.9559 ± 0.0027 | **0.9804 ± 0.0015** |
| | ICEWS1819 | ✗ | 0.9824 ± 0.0062 | 0.9813 ± 0.0029 | 0.9805 ± 0.0052 | 0.9838 ± 0.0031 | 0.9866 ± 0.0017 | 0.9610 ± 0.0085 | **0.9884 ± 0.0013** |
| | | ✓ | 0.9752 ± 0.0037 | 0.9676 ± 0.0026 | 0.9908 ± 0.0032 | 0.9886 ± 0.0025 | **0.9927 ± 0.0012** | 0.9871 ± 0.0034 | 0.9901 ± 0.0018 |
| | Googlemap CT | ✗ | OOM | OOM | 0.8480 ± 0.0026 | **0.8511 ± 0.0022** | 0.7789 ± 0.0014 | 0.6768 ± 0.0071 | 0.7865 ± 0.0023 |
| | | ✓ | OOM | OOM | **0.9002 ± 0.0019** | 0.8721 ± 0.0027 | 0.8335 ± 0.0018 | 0.8072 ± 0.0010 | 0.8183 ± 0.0038 |
| | GDELT | ✗ | 0.9482 ± 0.0018 | 0.9415 ± 0.0013 | 0.9342 ± 0.0039 | 0.9398 ± 0.0036 | 0.9554 ± 0.0006 | 0.9299 ± 0.0028 | **0.9640 ± 0.0002** |
| | | ✓ | 0.9466 ± 0.0032 | 0.9416 ± 0.0017 | 0.9572 ± 0.0029 | 0.9582 ± 0.0053 | 0.9601 ± 0.0011 | 0.9523 ± 0.0020 | **0.9653 ± 0.0003** |
| | Stack elec | ✗ | OOM | OOM | 0.9489 ± 0.0003 | 0.9496 ± 0.0031 | 0.8613 ± 0.0062 | 0.9502 ± 0.0011 | **0.9746 ± 0.0015** |
| | | ✓ | OOM | OOM | 0.9646 ± 0.0005 | 0.9529 ± 0.0023 | 0.9441 ± 0.0079 | 0.9591 ± 0.0009 | **0.9819 ± 0.0010** |
| | Stack ubuntu | ✗ | OOM | OOM | 0.9040 ± 0.0564 | **0.9481 ± 0.0007** | 0.9412 ± 0.0018 | 0.9417 ± 0.0016 | 0.9432 ± 0.0026 |
| | | ✓ | OOM | OOM | 0.9352 ± 0.0012 | 0.9420 ± 0.0005 | 0.9416 ± 0.0023 | 0.9416 ± 0.0047 | **0.9431 ± 0.0008** |
| | Amazon movies | ✗ | OOM | OOM | 0.8733 ± 0.0011 | 0.8502 ± 0.0003 | 0.8762 ± 0.0005 | **0.8776 ± 0.0002** | 0.8766 ± 0.0007 |
| | | ✓ | OOM | OOM | 0.9065 ± 0.0016 | 0.8885 ± 0.0007 | 0.9051 ± 0.0010 | 0.8906 ± 0.0006 | **0.9097 ± 0.0003** |
| | Yelp | ✗ | OOM | OOM | 0.8622 ± 0.0032 | 0.8396 ± 0.0018 | 0.8583 ± 0.0017 | 0.8472 ± 0.0010 | **0.8679 ± 0.0008** |
| | | ✓ | OOM | OOM | 0.9457 ± 0.0025 | 0.9323 ± 0.0027 | **0.9511 ± 0.0013** | 0.8883 ± 0.0026 | 0.9391 ± 0.0011 |
| *in.* | Enron | ✗ | 0.8635 ± 0.0018 | 0.8682 ± 0.0049 | 0.8180 ± 0.0034 | 0.8901 ± 0.0023 | 0.8433 ± 0.0043 | 0.7482 ± 0.0066 | **0.9199 ± 0.0021** |
| | | ✓ | 0.8761 ± 0.0023 | 0.7734 ± 0.0044 | 0.8589 ± 0.0031 | 0.9223 ± 0.0111 | 0.8560 ± 0.0024 | 0.8328 ± 0.0034 | **0.9409 ± 0.0025** |
| | ICEWS1819 | ✗ | 0.9382 ± 0.0071 | 0.9444 ± 0.0037 | 0.9181 ± 0.0056 | 0.9406 ± 0.0028 | 0.9536 ± 0.0018 | 0.8811 ± 0.0076 | **0.9665 ± 0.0011** |
| | | ✓ | 0.9333 ± 0.0026 | 0.9134 ± 0.0041 | 0.9716 ± 0.0033 | 0.9631 ± 0.0034 | **0.9789 ± 0.0022** | 0.9625 ± 0.0030 | 0.9688 ± 0.0018 |
| | Googlemap CT | ✗ | OOM | OOM | 0.7843 ± 0.0020 | **0.7942 ± 0.0024** | 0.7274 ± 0.0016 | 0.6624 ± 0.0060 | 0.7293 ± 0.0011 |
| | | ✓ | OOM | OOM | **0.8750 ± 0.0015** | 0.8012 ± 0.0021 | 0.7936 ± 0.0009 | 0.7633 ± 0.0013 | 0.7735 ± 0.0031 |
| | GDELT | ✗ | 0.9072 ± 0.0017 | 0.8756 ± 0.0022 | 0.7500 ± 0.0067 | 0.7980 ± 0.0016 | 0.8430 ± 0.0053 | 0.7298 ± 0.0060 | **0.9172 ± 0.0014** |
| | | ✓ | 0.9019 ± 0.0023 | 0.8928 ± 0.0011 | 0.9023 ± 0.0010 | 0.8986 ± 0.0077 | 0.9151 ± 0.0045 | 0.8925 ± 0.0048 | **0.9263 ± 0.0009** |
| | Stack elec | ✗ | OOM | OOM | 0.7784 ± 0.0043 | 0.7816 ± 0.0123 | 0.7346 ± 0.0068 | 0.7874 ± 0.0040 | **0.8742 ± 0.0061** |
| | | ✓ | OOM | OOM | 0.8391 ± 0.0036 | 0.7921 ± 0.0086 | 0.7599 ± 0.0041 | 0.8142 ± 0.0021 | **0.8801 ± 0.0043** |
| | Stack ubuntu | ✗ | OOM | OOM | 0.7365 ± 0.0421 | 0.7957 ± 0.0030 | 0.7746 ± 0.0058 | 0.7764 ± 0.0105 | **0.8054 ± 0.0027** |
| | | ✓ | OOM | OOM | 0.7664 ± 0.0015 | **0.7882 ± 0.0015** | 0.7777 ± 0.0015 | 0.7870 ± 0.0015 | 0.7832 ± 0.0015 |
| | Amazon movies | ✗ | OOM | OOM | 0.8322 ± 0.0015 | 0.7974 ± 0.0004 | 0.8311 ± 0.0015 | **0.8325 ± 0.0003** | 0.8324 ± 0.0004 |
| | | ✓ | OOM | OOM | 0.8760 ± 0.0010 | 0.8508 ± 0.0006 | 0.8738 ± 0.005 | 0.8517 ± 0.0007 | **0.8780 ± 0.0006** |
| | Yelp | ✗ | OOM | OOM | 0.8212 ± 0.0019 | 0.7942 ± 0.0007 | 0.8159 ± 0.0014 | 0.8028 ± 0.0008 | **0.8251 ± 0.0015** |
| | | ✓ | OOM | OOM | 0.9174 ± 0.0010 | 0.9010 ± 0.0009 | **0.9249 ± 0.0013** | 0.8494 ± 0.0008 | 0.9092 ± 0.0006 |

Table 11: Hits@1 for destination node retrieval. *tr.* means transductive setting and *in.* means inductive setting. **Text** means whether to use Bert-encoded embeddings for initialization.

| | Datasets | Text | JODIE | DyRep | TGAT | CAWN | TCL | GraphMixer | DyGFormer |
|---|---|---|---|---|---|---|---|---|---|
| *tr.* | Enron | ✗ | 0.4610 ± 0.0078 | OOM | 0.2649 ± 0.0039 | 0.0530 ± 0.0041 | 0.3083 ± 0.0023 | 0.3107 ± 0.0022 | **0.4928 ± 0.0011** |
| | | ✓ | 0.5045 ± 0.0083 | OOM | 0.5366 ± 0.0026 | 0.6590 ± 0.0018 | 0.4344 ± 0.0020 | 0.4405 ± 0.0015 | **0.7473 ± 0.0023** |
| | ICEWS1819 | ✗ | 0.6437 ± 0.0017 | 0.6329 ± 0.0024 | 0.4523 ± 0.0053 | 0.4755 ± 0.0049 | 0.5142 ± 0.0033 | 0.5288 ± 0.0016 | **0.6630 ± 0.0013** |
| | | ✓ | 0.6603 ± 0.0010 | 0.6133 ± 0.0026 | 0.7809 ± 0.0031 | 0.7812 ± 0.0051 | **0.8188 ± 0.0121** | 0.8003 ± 0.0078 | 0.8036 ± 0.0002 |
| | Googlemap CT | ✗ | OOM | OOM | 0.0127 ± 0.0065 | 0.0122 ± 0.0034 | **0.0156 ± 0.0002** | 0.0110 ± 0.0001 | 0.0133 ± 0.0005 |
| | | ✓ | OOM | OOM | **0.2290 ± 0.0042** | 0.1322 ± 0.0019 | 0.1468 ± 0.0004 | 0.1202 ± 0.0005 | 0.1314 ± 0.0002 |
| | GDELT | ✗ | 0.3323 ± 0.0027 | 0.3127 ± 0.0015 | 0.2453 ± 0.0017 | 0.2177 ± 0.0082 | 0.2083 ± 0.0037 | 0.0611 ± 0.0052 | **0.3556 ± 0.0029** |
| | | ✓ | 0.2890 ± 0.0031 | 0.3112 ± 0.0029 | 0.4135 ± 0.0013 | 0.4253 ± 0.0010 | 0.4353 ± 0.0024 | 0.3875 ± 0.0033 | **0.4764 ± 0.0008** |
| | Amazon movies | ✗ | OOM | OOM | 0.2407 ± 0.0013 | 0.1678 ± 0.0023 | 0.2373 ± 0.0014 | **0.2504 ± 0.0007** | 0.2139 ± 0.0009 |
| | | ✓ | OOM | OOM | 0.2851 ± 0.0009 | 0.2237 ± 0.0026 | 0.2841 ± 0.0013 | 0.2613 ± 0.0008 | **0.3008 ± 0.0007** |
| | Yelp | ✗ | OOM | OOM | **0.1774 ± 0.0039** | 0.1473 ± 0.0024 | 0.1747 ± 0.0038 | 0.1554 ± 0.0031 | 0.1284 ± 0.0017 |
| | | ✓ | OOM | OOM | **0.3737 ± 0.0051** | 0.3218 ± 0.0104 | 0.1554 ± 0.0028 | 0.2204 ± 0.0037 | 0.3512 ± 0.0027 |
| *in.* | Enron | ✗ | 0.2719 ± 0.0055 | OOM | 0.1301 ± 0.0027 | 0.0391 ± 0.0016 | 0.1473 ± 0.0025 | 0.1518 ± 0.0013 | **0.2964 ± 0.0019** |
| | | ✓ | 0.3003 ± 0.0042 | OOM | 0.2832 ± 0.0033 | 0.4223 ± 0.0029 | 0.1967 ± 0.0014 | 0.2046 ± 0.0009 | **0.5553 ± 0.0027** |
| | ICEWS1819 | ✗ | 0.4834 ± 0.0126 | 0.4657 ± 0.0084 | 0.2209 ± 0.0076 | 0.4077 ± 0.0008 | 0.2883 ± 0.0145 | 0.2979 ± 0.0023 | **0.5009 ± 0.0073** |
| | | ✓ | 0.5134 ± 0.0010 | 0.4799 ± 0.0026 | 0.5752 ± 0.0031 | 0.6342 ± 0.0051 | 0.6059 ± 0.0122 | 0.6114 ± 0.0078 | **0.634 ± 0.00028** |
| | Googlemap CT | ✗ | OOM | OOM | 0.0109 ± 0.0007 | 0.0105 ± 0.0003 | 0.0118 ± 0.0005 | **0.0170 ± 0.0001** | 0.0055 ± 0.0002 |
| | | ✓ | OOM | OOM | **0.0550 ± 0.0007** | 0.0321 ± 0.0004 | 0.0236 ± 0.0005 | 0.0191 ± 0.0001 | 0.0178 ± 0.0006 |
| | GDELT | ✗ | **0.3040 ± 0.0061** | 0.2678 ± 0.0088 | 0.1270 ± 0.0056 | 0.2138 ± 0.0043 | 0.1854 ± 0.0091 | 0.0241 ± 0.0036 | 0.2092 ± 0.0027 |
| | | ✓ | 0.3080 ± 0.0074 | 0.2740 ± 0.0051 | 0.2630 ± 0.0042 | 0.2630 ± 0.0011 | 0.2916 ± 0.0031 | 0.2610 ± 0.0046 | **0.3681 ± 0.0038** |
| | Amazon movies | ✗ | OOM | OOM | **0.2087 ± 0.0037** | 0.1406 ± 0.0017 | 0.2071 ± 0.0015 | 0.1906 ± 0.0004 | 0.1886 ± 0.0038 |
| | | ✓ | OOM | OOM | 0.2403 ± 0.0038 | 0.1835 ± 0.0042 | 0.2463 ± 0.0008 | 0.2224 ± 0.0009 | **0.2613 ± 0.0003** |
| | Yelp | ✗ | OOM | OOM | **0.1515 ± 0.0105** | 0.1240 ± 0.0087 | 0.1490 ± 0.0108 | 0.1383 ± 0.0054 | 0.0984 ± 0.0042 |
| | | ✓ | OOM | OOM | **0.3086 ± 0.0238** | 0.2642 ± 0.0071 | 0.1383 ± 0.0105 | 0.1841 ± 0.0014 | 0.2898 ± 0.0036 |

#### C.4.4 Textual Relation Generation

As shown in Table 13, we provide the performance of textual relation generation task on more datasets. As shown in Figure 12, the influence of the input history length is studied for the relation generation performance. We report the performance of Vicuna-7b on the `Googlemap CT` dataset and the performance of Llama3-8b on the `Stack elec` dataset. We control the history length by sampling the recent $k$ reviews. We have two observations. First, the performance may degrade with a larger history length (*e.g.,* precision on `Stack elec`). This observation shows the necessity of designing strategies to flexibly handle history text for different samples. Second, the fine-tuning process can stabilize the performance of large language models when facing long text history (*e.g.,*

Table 12: Hits@3 for destination node retrieval. *tr.* means transductive setting and *in.* means inductive setting. **Text** means whether to use Bert-encoded embeddings for initialization.

| | Datasets | Text | JODIE | DyRep | TGAT | CAWN | TCL | GraphMixer | DyGFormer |
|---|---|---|---|---|---|---|---|---|---|
| *tr.* | Enron | ✗ | **0.7297 ± 0.0412** | OOM | 0.5103 ± 0.0023 | 0.3084 ± 0.0037 | 0.5085 ± 0.0049 | 0.5105 ± 0.0062 | 0.6645 ± 0.0041 |
| | | ✓ | 0.7612 ± 0.0311 | OOM | 0.7583 ± 0.0082 | 0.7711 ± 0.0064 | 0.7008 ± 0.0081 | 0.6754 ± 0.0027 | **0.8877 ± 0.0051** |
| | ICEWS1819 | ✗ | 0.6437 ± 0.0024 | 0.6016 ± 0.0018 | 0.6512 ± 0.0027 | 0.6836 ± 0.0019 | 0.7111 ± 0.0039 | 0.7094 ± 0.0046 | **0.8192 ± 0.0021** |
| | | ✓ | 0.8532 ± 0.0012 | 0.8415 ± 0.0061 | 0.9188 ± 0.0034 | 0.8951 ± 0.0078 | **0.9356 ± 0.0106** | 0.9231 ± 0.0013 | 0.9175 ± 0.0006 |
| | Googlemap CT | ✗ | OOM | OOM | **0.2579 ± 0.0058** | 0.1824 ± 0.0007 | 0.1750 ± 0.0004 | 0.1743 ± 0.0004 | 0.2556 ± 0.0008 |
| | | ✓ | OOM | OOM | **0.4325 ± 0.0023** | 0.2771 ± 0.0018 | 0.2966 ± 0.0031 | 0.2514 ± 0.0028 | 0.2705 ± 0.0021 |
| | GDELT | ✗ | **0.6184 ± 0.0042** | 0.5694 ± 0.0067 | 0.3052 ± 0.0072 | 0.2649 ± 0.0053 | 0.4560 ± 0.0080 | 0.1585 ± 0.0076 | 0.6130 ± 0.0013 |
| | | ✓ | 0.5752 ± 0.0035 | 0.5731 ± 0.0028 | 0.6621 ± 0.0019 | 0.6631 ± 0.0017 | 0.6756 ± 0.0029 | 0.6351 ± 0.0030 | **0.7089 ± 0.0010** |
| | Amazon movies | ✗ | OOM | OOM | 0.4128 ± 0.0038 | 0.3189 ± 0.0039 | 0.4133 ± 0.0053 | **0.4199 ± 0.0013** | 0.3506 ± 0.0013 |
| | | ✓ | OOM | OOM | 0.4855 ± 0.0013 | 0.4122 ± 0.0065 | 0.4800 ± 0.0011 | 0.4438 ± 0.0013 | **0.4990 ± 0.0008** |
| | Yelp | ✗ | OOM | OOM | **0.3471 ± 0.0064** | 0.2876 ± 0.0032 | 0.3457 ± 0.0126 | 0.2986 ± 0.0128 | 0.2686 ± 0.0103 |
| | | ✓ | OOM | OOM | **0.5968 ± 0.0043** | 0.5393 ± 0.0024 | 0.2986 ± 0.0084 | 0.4017 ± 0.0042 | 0.5549 ± 0.0054 |
| *in.* | Enron | ✗ | **0.4987 ± 0.0029** | OOM | 0.2635 ± 0.0035 | 0.2855 ± 0.0042 | 0.2724 ± 0.0047 | 0.2691 ± 0.0019 | 0.4608 ± 0.0082 |
| | | ✓ | 0.5244 ± 0.0016 | OOM | 0.5045 ± 0.0025 | 0.5962 ± 0.0027 | 0.3928 ± 0.0035 | 0.3916 ± 0.0011 | **0.7474 ± 0.0026** |
| | ICEWS1819 | ✗ | **0.7035 ± 0.0125** | 0.6673 ± 0.0193 | 0.4148 ± 0.0032 | 0.4723 ± 0.0079 | 0.4663 ± 0.0128 | 0.4417 ± 0.0113 | 0.6748 ± 0.0087 |
| | | ✓ | 0.7140 ± 0.0086 | 0.7010 ± 0.0032 | 0.7835 ± 0.0029 | **0.8120 ± 0.0066** | 0.8026 ± 0.0091 | 0.8001 ± 0.0088 | 0.8017 ± 0.0071 |
| | Googlemap CT | ✗ | OOM | OOM | 0.0308 ± 0.0003 | 0.0292 ± 0.0009 | **0.0352 ± 0.0012** | 0.0346 ± 0.0001 | 0.0174 ± 0.0016 |
| | | ✓ | OOM | OOM | **0.1545 ± 0.0017** | 0.0828 ± 0.0021 | 0.0621 ± 0.0008 | 0.0465 ± 0.0004 | 0.0448 ± 0.0007 |
| | GDELT | ✗ | **0.5352 ± 0.0011** | 0.4940 ± 0.0059 | 0.2632 ± 0.0072 | 0.2929 ± 0.0043 | 0.3434 ± 0.0011 | 0.0699 ± 0.0077 | 0.4341 ± 0.0039 |
| | | ✓ | 0.5294 ± 0.0028 | 0.5179 ± 0.0068 | 0.4818 ± 0.0055 | 0.4667 ± 0.0061 | 0.5090 ± 0.0024 | 0.4755 ± 0.0081 | **0.5666 ± 0.0026** |
| | Amazon movies | ✗ | OOM | OOM | 0.3511 ± 0.0087 | 0.2634 ± 0.0033 | 0.3496 ± 0.0028 | **0.3563 ± 0.0026** | 0.3112 ± 0.0079 |
| | | ✓ | OOM | OOM | 0.4165 ± 0.0061 | 0.3441 ± 0.0051 | 0.4157 ± 0.0023 | 0.3764 ± 0.0011 | **0.4319 ± 0.0011** |
| | Yelp | ✗ | OOM | OOM | **0.3009 ± 0.0241** | 0.2420 ± 0.0064 | 0.2958 ± 0.0154 | 0.2594 ± 0.0069 | 0.2157 ± 0.0051 |
| | | ✓ | OOM | OOM | **0.5120 ± 0.0206** | 0.4553 ± 0.0120 | 0.2594 ± 0.0168 | 0.3413 ± 0.0060 | 0.4751 ± 0.0064 |

Table 13: Precision, Recall and F1 of BERTscore of different LLMs for the textural relation generation tasks. The number of test samples is 500 per dataset.

| | ICEWS1819 | | | Enron | | | Stack ubuntu | | |
|---|---|---|---|---|---|---|---|---|---|
| | Precision | Recall | F1 | Precision | Recall | F1 | Precision | Recall | F1 |
| Llama3-8b | 77.38 | 82.23 | 79.71 | 79.88 | 79.26 | 79.46 | 79.48 | 82.86 | 81.10 |
| Mistral-7b | 78.15 | **82.52** | 80.25 | 79.75 | **79.37** | 79.52 | 79.61 | 82.98 | 81.24 |
| Vicuna-7b | 77.44 | 82.17 | 79.71 | **81.50** | 78.62 | **79.98** | 80.92 | **83.01** | 81.93 |
| Vicuna-13b | **78.81** | 82.38 | **80.51** | 80.74 | 78.99 | 79.81 | 81.39 | 82.99 | **82.15** |

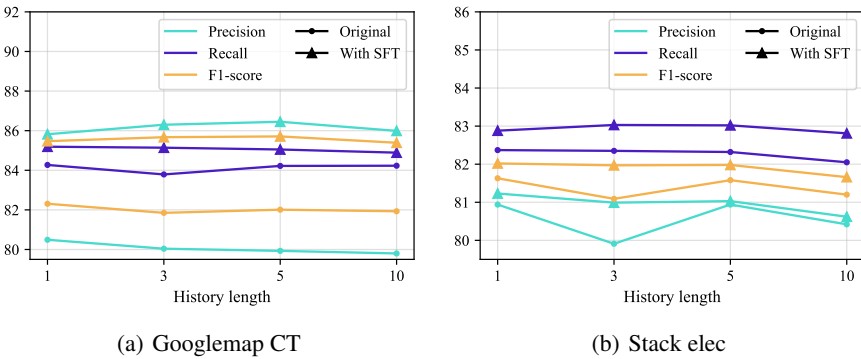

(a) Googlemap CT                (b) Stack elec

Figure 12: Textual relation generation performance with different history lengths.

precision and F1-score on `Stack elec`). This shows the effectiveness of supervised fine-tuning in enhancing the ability of LLMs to understand sequential interaction contexts.

