# OpenReview forum: "DTGB: A Comprehensive Benchmark for Dynamic Text-Attributed Graphs"
_NeurIPS.cc/2024/Datasets_and_Benchmarks_Track — NeurIPS 2024 Track Datasets and Benchmarks Poster_

### Official Review · Reviewer_obHN · 2024-07-20
**Review of Submission2182**

**Rating:** 9
**Confidence:** 5
**Correctness:** Yes.
**Clarity:** Yes.

**Review:**

Please refer to "Strengths" and "Opportunities For Improvement" for details.

**Strengths:**

1.	The paper introduces the first benchmark for dynamic text-attributed graphs, providing a new standard for evaluating models.
2.	The paper includes a variety of large-scale, domain-diverse datasets with rich text attributes for dynamic graphs.
3.	The paper establishes a standardized evaluation framework with four distinct tasks: future link prediction, destination node retrieval, edge classification, and textual relation generation.

**Additional Feedback:**

N/A

**Documentation:**

Yes.

**Ethics:**

No.

**Limitations:**

Yes.

**Opportunities For Improvement:**

1.	In Table 2, all baselines have low accuracy on GDELT and ICEWS1819. Is it because the edges of these two datasets are inherently difficult to distinguish?
2.	Textural relation generation is an interesting task and the results in Table 5 are quite good. Is it due to the powerful generation ability of GPT-4o?
3.	The order of downstream tasks, which are presented in the experimental section, should be consistent with that in Section 4.
4.	The basic paradigm of the DyTAG’s pre-trained model should be explained in advance, which will help readers follow.

**Relation To Prior Work:**

Yes.

**Summary And Contributions:**

This paper addresses the gap in benchmark datasets for DyTAG datasets and evaluates them on four challenge downstream tasks: future link prediction, destination node retrieval, edge classification, and textual relation generation.

---

> ### Author Rebuttal · Authors · 2024-08-17
>
> Thanks for your valuable feedback and positive comments on our dataset from diverse domains and comprehensive evaluation.  We address your potential concerns as follows.
>
> >Q1: In Table 2, all baselines have low accuracy on GDELT and ICEWS1819. Is it because the edges of these two datasets are inherently difficult to distinguish?
>
> The lower accuracy across all baselines for GDELT and ICEWS1819 stems from two key factors that make these datasets particularly challenging.
> Firstly, as evident from Table 1, GDELT and ICEWS1819 possess **a significantly larger number of edge types** compared to the other datasets in our benchmark. This high diversity of relationships substantially increases the complexity of the edge prediction task.
> Secondly, we've identified that most **existing models lack sophisticated modeling to process rich edge features**. As GDELT and ICEWS1819 datasets contain complex, nuanced relationships between entities that are encoded in detailed edge attributes, the insufficient edge feature modeling leads to worse performance on these two datasets.
>
>
> >Q2: Textural relation generation is an interesting task and the results in Table 5 are quite good. Is it due to the powerful generation ability of GPT-4o?
>
> Yes. The powerful generation ability of LLMs, such as GPT-4o, plays a significant role in the good results in Table 5. We also would like to clarify that the strong performance stems from multiple key factors beyond just leveraging a powerful LLM:
> **Tailored Prompt and Graph Construction:** We develop a sophisticated prompting strategy that incorporates the user's interaction history and preferences. This enables LLMs to better understand and reason about the relationships between entities.
> **Domain-Specific Fine-tuning:** We implement supervised fine-tuning to LLMs, so as to adapt to domain-specific datasets. This further enhances the performance on this particular task.
> Our carefully curated dataset and comprehensive testing with LLMs not only demonstrate current capabilities but also provide valuable insights and directions for future advancements in textual relation generation models.
>
> >Q3: The order of downstream tasks, which are presented in the experimental section, should be consistent with that in Section 4.
>
> Thanks for your suggestions. We will adjust the order accordingly in our revised version for better readability.
>
> >Q4: The basic paradigm of the DyTAG’s pre-trained model should be explained in advance, which will help readers follow.
>
> For future link prediction and edge classification, we use the traditional training and evaluation paradigm. For the destination node retrieval task, we discussed the training and evaluation paradigm in **Appendix C.1 (Line 624-626)**. Thanks for your suggestions. We will add more discussion on the pre-training paradigm in our revised version for better readability.

---

### Official Review · Reviewer_gCk2 · 2024-07-24
**Review for DTGB: A Comprehensive Benchmark for Dynamic Text-Attributed Graphs**

**Rating:** 7
**Confidence:** 3
**Correctness:** The experiments are done in an approp…

**Review:**

This paper proposes eight datasets tailored for dynamic text-attributed graphs, along with four tasks featuring standardized evaluation protocols to assess graph algorithms and LLMs. The benchmark contributes significantly to the field of DyTAGs. However, the paper does not include newly proposed algorithms that integrate GNNs with LLMs, which may limit the applicability and impact of the benchmark.

**Strengths:**

1. Eight dynamic text-attributed graphs are collected from various domains, addressing the lack of benchmark datasets in the field.
2. The paper designs four tasks to evaluate the performance of dynamic graph models and LLMs that require models to understand both dynamic graph structures and natural language.
3. This paper is clearly written and easy to follow.

**Additional Feedback:**

The DyTAGs in the paper have all been processed in the form of discrete-time dynamic graphs. I am curious about whether continuous-time dynamic graphs can be considered.

**Clarity:**

This paper is clearly written and easy to follow.

Minor points:
1. "Ours" in Table 1 is not centered.
2. There is a issue with the header 'Models' in Table 2.

**Documentation:**

Experimental setup and code are provided and documented well.

**Ethics:**

There are no ethical concerns.

**Limitations:**

Limitations are fully discussed in the conclusion.

**Opportunities For Improvement:**

The tasks proposed by DTGB evaluate the effectiveness of dynamic graph learning models and the ability of LLMs to understand dynamic graphs independently. There are already some works combining LLMs and GNNs [1]. I wonder if these approaches can also be incorporated into DTGB.

[1] Ren et al. A Survey of Large Language Models for Graphs. KDD 2024.

**Relation To Prior Work:**

Related works are adequately discussed and appropriately cited.

**Summary And Contributions:**

This paper proposes a comprehensive benchmark DTGB for dynamic text-attributed graphs. The authors collect 8 datasets and design four tasks to evaluate 7 dynamic graph learning algorithms and 6 powerful large language models. Extensive experiments are conducted to demonstrate the utility of DTGB.

---

> ### Author Rebuttal · Authors · 2024-08-17
>
> Thanks for your valuable feedback and positive comments on paper writing, dataset contribution, and comprehensive evaluation.  We address your potential concerns as follows.
>
> >Q1: The paper does not include newly proposed algorithms that integrate GNNs with LLMs, which may limit the applicability and impact of the benchmark. There are already some works combining LLMs and GNNs. I wonder if these approaches can also be incorporated into DTGB.
>
> We appreciate your valuable suggestion regarding the integration of GNNs with LLMs. We have briefly discussed this aspect in the limitation and future direction part of the manuscript **(Line 795-801 in Appendix D): "A particularly exciting future direction is the design of temporal graph tokens that can directly incorporate dynamic graph information into LLMs for reasoning and dynamics-aware generation..."**. We acknowledge the potential benefits of existing  LLM+GNN approaches, but they **typically overlook the crucial dynamic inherent in our dataset** and **lack temporal modeling on historical interactions**. Therefore, adapting existing LLM+GNN models to fully leverage this dynamic textual information would require substantial research efforts and algorithm development, which we believe warrants a separate, dedicated study.
>
> Given the scope and limited timeline of the rebuttal period, we focus on establishing a comprehensive baseline for dynamic textual graph benchmarking in this work. We anticipate that researchers will build upon our findings to develop more sophisticated models that can fully utilize both the structural and temporal-textual aspects of dynamic graphs.
>
> We agree that a more detailed analysis of the limitations of existing LLM+GNN methods and the challenges of integrating temporal graph information into LLMs could better inspire future research. Accordingly, we will expand our discussion on these aspects in the revised version.
>
> >Q2: "Ours" in Table 1 is not centered. / There is an issue with the header 'Models' in Table 2.
>
> Thanks for pointing this out. We have addressed the formatting issue in our revised version.

---

> > ### Comment · Reviewer_gCk2 · 2024-08-24
> >
> > Thanks for the detailed response. My concern has been solved. I will keep my positive rating.

---

### Official Review · Reviewer_FWvA · 2024-07-25

**Rating:** 7
**Confidence:** 4
**Correctness:** Yes
**Clarity:** Yes

**Review:**

This work is well organized and innovates in the new datasets and corresponding benchmark experiments. Specifically, the pros and cons are listed below.

Pros:
1. The paper introduces the first open benchmark, including 8 DyTAG datasets, for dynamic text-attributed graphs.
2. Building upon the DyTAG datasets, they develop a standardized evaluation protocol to facilitation a comprehensive and fair evaluation of existing algorithms.
3. Interesting observations are obtained by extensive benchmark experiments across various evaluation settings. These results further demonstrate the importance of mining the textual attributed associated with dynamic graphs.

Cons.
1. While extensive experiments are conducted in this work, it would be better to see some results regarding few-shot learning since the current supervised setting may be not enough to assess the capabilities of LLM- and Graph-based models.

**Strengths:**

1. The proposed DTGB is interesting to a broad of researcher in graph machine learning.
2. The authors provide a comprehensive and easy to use standardized protocol for researchers interested in this topic.

**Additional Feedback:**

Please see the comments above.

**Documentation:**

Yes

**Limitations:**

Yes

**Opportunities For Improvement:**

Please see the comments above.

**Relation To Prior Work:**

Yes

**Summary And Contributions:**

This paper focuses on dynamic text-attributed graphs (DyTAGs) and introduces DTGB, a collection of large-scale, time-evolving graphs from diverse domains. Additionally, the authors also conduct extensive benchmark experiments on DTGB, providing insights on the limitations of existing models and demonstrating the utility of DTGB in investigating the incorporation of structural and textual dynamics.

---

> ### Author Rebuttal · Authors · 2024-08-17
>
> Thanks for your valuable feedback and positive comments on dataset innovation, comprehensive evaluation and insightful observations.  We address your potential concerns as follows.
>
> >Q1: It would be better to see some results regarding few-shot learning since the current supervised setting may be not enough to assess the capabilities of LLM- and Graph-based models.
>
> We would like to emphasize that our current benchmark design, particularly for Temporal Graph Neural Networks (TGNNs), already incorporates the few-shot experiments that address your concern.
>
> In our Future Link Prediction and Destination Node Retrieval tasks,  **we have implemented an inductive setting**. In these tasks, TGNNs are required to make predictions on previously unseen nodes or edges, effectively testing their ability to generalize from limited information. This design **inherently presents a form of few-shot learning scenario for TGNNs**. From Table 3 and Table 9, we can see that **text attributes consistently help existing dynamic graph models in the inductive setting**.  For example, with text attributes, GraphMixer achieves **11.0% higher AUC-ROC** on the Enron dataset, and TGAT achieves **16.4% higher Hits@10** on the Amazon movies dataset. Demonstrating the effectiveness of text attributes in improving the generalization ability of existing dynamic graph models.
>
> For textual relation generation, we further implemented few-shot experiments and reported the F-1 score of different settings as follows. The k-shot indicates providing the LLM with k examples of task instructions.
>
> | BERTScore-F1 (Googlemap CT) | 0-shot | 1-shot | 5-shot | 10-shot |
> | :----- | :--: | :--: |  :--: |  :--: |
> | Vicuna-7b |81.85 |	82.11 |	82.28 |	82.32
> | Vicuna-13b |81.99 |	81.90 |	82.26 |	82.39
>
> | BERTScore-F1 (Stack elec) | 0-shot | 1-shot | 5-shot | 10-shot |
> | :----- | :--: | :--: |  :--: |  :--: |
> | Vicuna-7b |81.46 |	81.57 | 81.70 |	82.18
> | Vicuna-13b |81.33 | 82.53 | 82.63 | 82.69

---

### Official Review · Reviewer_GM9U · 2024-07-25
**This paper proposes a dynamic text-attributed graph benchmark, which consists of time-evolving graphs from diverse domains.**

**Rating:** 7
**Confidence:** 4
**Clarity:** Yes

**Review:**

Pros:

1. The authors have collected 8 dynamic text-attributed graphs where both nodes and edges possess text attributes.
2. The authors evaluated the exiting dynamic graph learning models on the proposed datasets with four downstream tasks.
3. The paper is well-written and easy to follow.

Cons:

1. The authors claim that "DTGB distinguishes itself with its rich text, long-range historical information, and large-scale dynamic structures". However, there is no evidence demonstrating the effect of long-range historical information and dynamic structure. For example, if the test set is randomly sampled instead of using the timestamps, does the performance show significant differences?
2. From the experimental results of Edge Classification and Link Prediction, it seems the text attributes are not helpful.

**Strengths:**

Please refer to the Review.

**Additional Feedback:**

N/A

**Correctness:**

The datasets are constructed in a sound way and the experimental designs are appropriate.

**Documentation:**

Yes

**Opportunities For Improvement:**

This work can be improved by demonstrating the effect of long-range historical information and dynamic structure of the proposed datasets.

**Relation To Prior Work:**

Yes

**Summary And Contributions:**

This paper introduces a dynamic text-attributed graph benchmark (DTGB), featuring time-evolving graphs from various domains. Specifically, the authors have collected 8 DyTAG datasets where both nodes and edges possess text attributes. Additionally, they assess existing models on these datasets across four downstream tasks.

---

> ### Author Rebuttal · Authors · 2024-08-17
>
> Thanks for your valuable feedback and positive comments on paper writing, dataset contribution, and diverse benchmark.  We address your potential concerns as follows.
>
> >Q1: There is no evidence demonstrating the effect of long-range historical information and dynamic structure. For example, if the test set is randomly sampled instead of using the timestamps, does the performance show significant differences?
>
> Thanks for your suggestions. To demonstrate the effect of dynamic structure, we conducted two ablation studies on the future link prediction task. First, we randomly shuffled the historical timestamps, **disturbing the chronological order of historical links**. Second, we removed the timestamps from historical interactions, **treating the history as a static graph**. As shown in the tables below, both the shuffled historical structure and the static historical structure **resulted in significant performance degradation**, highlighting that the accurate dynamic structure is essential for the performance of dynamic graph models.
>
> | AUC-ROC (ICEWS1819) | TGAT | CAWN | TCL | GraphMixer | GraphFormer |
> | :----- | :--: | :--: |  :--: |  :--: | :--: |
> | Shuffled historical structure |0.8810 |	0.9039 |	0.8975 |	0.8807 |	0.8712
> | Static historical structure |0.8931 |	0.9217 |	0.9244 |	0.8792 |	0.8936
> | Original |0.9904 |	0.9857 |	0.9923 |	0.9863 |	0.9888
>
> | AUC-ROC (Enron) | TGAT | CAWN | TCL | GraphMixer | GraphFormer |
> | :----- | :--: | :--: |  :--: |  :--: | :--: |
> | Shuffled historical structure |0.7965 |	0.8079 |	0.6733 |	0.8314 |	0.8447
> | Static historical structure |0.8133 |	0.8251 |	0.6931 |	0.8325 |	0.8864
> | Original |0.9681 |	0.9740 |	0.9618 |	0.9567 |	0.9779
>
> To demonstrate the effect of long-range history, we evaluate the future link prediction performance of dynamic graph models with varying history lengths. As shown in the tables below, **extending the history consistently enhances model performance**, highlighting the importance of incorporating long-range history.
>
> | AUC-ROC (ICEWS1819) | TGAT | CAWN | TCL | GraphMixer | GraphFormer |
> | :----- | :--: | :--: |  :--: |  :--: | :--: |
> | 1 |0.9684 |	0.9515 |	0.9648 |	0.9633 |	0.9663
> | 10 |0.9825 |	0.9651 |	0.9789 |	0.9861 |	0.9791
> | 50 |0.9901 |	0.9834 |	0.9894 |	0.9838 |	0.9846
> | 100 |0.9903 |	0.9860 |	0.9914 |	0.9866 |	0.9883
>
> | AUC-ROC (Enron) | TGAT | CAWN | TCL | GraphMixer | GraphFormer |
> | :----- | :--: | :--: |  :--: |  :--: | :--: |
> | 1 |0.8361 |	0.8549 |	0.8655 |	0.8561 |	0.9091
> | 10 |0.9355 |	0.9502 |	0.9399 |	0.9353 |	0.9467
> | 50 |0.9604 |	0.9698 |	0.9579 |	0.9524 |	0.9718
> | 100 |0.9670 |	0.9750 |	0.9626 |	0.9439 |	0.9776
>
>
> >Q2: From the experimental results of Edge Classification and Link Prediction, it seems the text attributes are not helpful.
>
> As shown in Figures 4 and 10, incorporating text attributes significantly enhances the edge classification performance of existing dynamic graph models (e.g., TGAT, CAWN, TCL, GraphMixer, and DyGFormer) across all eight datasets. Notably, for CAWN, this integration leads to a **15.8%** improvement on the Yelp dataset and a **22.5%** improvement on the ICEWS1819 dataset.
>
> Tables 3 and 9 demonstrate that text attributes consistently boost the performance of these models on future link prediction tasks, particularly in challenging inductive settings where test nodes are not seen during training. For instance, with text attributes, GraphMixer achieves an **11.0%** higher AUC-ROC on the Enron dataset in the inductive setting, while TCL shows a **13.3%** increase in Average Precision on the Yelp dataset.
>
> As discussed in Lines 238-243 of the manuscript, **only memory-based models, JODIE and DyRep, get performance degradation when incorporating text attributes**. This occurs because these models update node representations incrementally based on the numerical attributes of edges, and the Bert-encoded initialization of edges will potentially mislead the update process. These experimental results demonstrate the effectiveness of integrating text attributes in enhancing various dynamic graph tasks, highlighting the importance of developing advanced models capable of handling the complex interplay between dynamic graph structures and natural language.

---

> > ### Comment · Reviewer_GM9U · 2024-08-20
> >
> > Thanks to the author for the detailed response. I will keep my positive rating.

---

### Author Rebuttal · Authors · 2024-08-17

Summary of Revision:

We sincerely thank all the reviewers for their valuable comments, which are instructive for us to improve our paper.

The reviewers generally expressed positive opinions about our paper. They noted that the proposed datasets are “**constructed in a sound way**”, “**interesting to a broad range of researchers in graph machine learning**”, and “**address the lack of benchmark datasets in the field**”.     Additionally, we “**provide a comprehensive and easy-to-use standardized protocol for researchers interested in this topic**”.     Our experiments were “**appropriately designed**” and “**conducted in an appropriate, controlled, and fair manner**”.     The experimental setup, code, and datasets were “**well-documented**” and the paper was “**clearly written and easy to follow**” and “**well-organized**”.

The reviewers also raised insightful and constructive concerns, which we have addressed through several key efforts:

- We conducted new ablation studies to clarify the effect of dynamic structure and long-range history in dynamic graph modeling (R1-Q1).
- We clarified the effectiveness of integrating text attributes (R1-Q2) and provided detailed explanations for the experimental results of concern (R4-Q1 and R4-Q2).
- We included new experimental results on textual relation generation in the few-shot setting to further explore this aspect of our datasets (R2-Q1).
- We emphasized the limitations and future directions discussed in our manuscript and elaborated on the inapplicability of existing GNN+LLM methods in our scenario (R3-Q1).

---

### Decision · Program_Chairs · 2024-09-26

**Decision:**

Accept (Poster)

**Comment:**

This paper introduces a dynamic text-attributed graph benchmark (DTGB), featuring time-evolving graphs from various domains. Specifically, the authors have collected 8 DyTAG datasets where both nodes and edges possess text attributes. Additionally, they assess existing models on these datasets across four downstream tasks.
All reviewers think it is a good paper and agree to accept it. So I suggest to accept it.